# Molecular basis of microhomology-mediated end-joining by purified full-length Polθ

Samuel J. Black [1], Ahmet Y. Ozdemir[1], Ekaterina Kashkina[1], Tatiana Kent[1], Timur Rusanov[1], Dejan Ristic[2], Yeonoh Shin[3], Antonio Suma [4], Trung Hoang[1], Gurushankar Chandramouly[1], Labiba A. Siddique[1], Nikita Borisonnik[1], Katherine Sullivan-Reed[5], Joseph S. Mallon[1], Tomasz Skorski [5], Vincenzo Carnevale [4], Katsuhiko S. Murakami [3], Claire Wyman[2] & Richard T. Pomerantz[1]

DNA polymerase θ (Polθ) is a unique polymerase-helicase fusion protein that promotes microhomology-mediated end-joining (MMEJ) of DNA double-strand breaks (DSBs). How full-length human Polθ performs MMEJ at the molecular level remains unknown. Using a biochemical approach, we find that the helicase is essential for Polθ MMEJ of long ssDNA overhangs which model resected DSBs. Remarkably, Polθ MMEJ of ssDNA overhangs requires polymerase-helicase attachment, but not the disordered central domain, and occurs independently of helicase ATPase activity. Using single-particle microscopy and biophysical methods, we find that polymerase-helicase attachment promotes multimeric gel-like Polθ complexes that facilitate DNA accumulation, DNA synapsis, and MMEJ. We further find that the central domain regulates Polθ multimerization and governs its DNA substrate requirements for MMEJ. These studies identify unexpected functions for the helicase and central domain and demonstrate the importance of polymerase-helicase tethering in MMEJ and the structural organization of Polθ.

[1] Fels Institute for Cancer Research, Department of Medical Genetics and Molecular Biochemistry, Temple University Lewis Katz School of Medicine, Philadelphia, PA 19140, USA. [2] Department of Molecular Genetics and Department of Radiation Oncology, Erasmus University Medical Center, 3000 CA Rotterdam, The Netherlands. [3] Department of Biochemistry and Molecular Biology, The Center for RNA Molecular Biology, Pennsylvania State University, University Park, PA 16802, USA. [4] Institute for Computational Molecular Science, Temple University, Philadelphia, PA, USA. [5] Fels Institute for Cancer Research, Department of Microbiology and Immunology, Temple University Lewis Katz School of Medicine, Philadelphia, PA 19140, USA. Correspondence and requests for materials should be addressed to R.T.P. (email: richard.pomerantz@temple.edu)

DNA polymerase θ (Polθ) is a polymerase-helicase fusion protein (Fig. 1a), that is essential for the double-strand break (DSB) repair pathway called microhomology-mediated end-joining (MMEJ) or alternative end-joining[1–6]. MMEJ functions during S and G2 cell-cycle phases and therefore acts on 3′ single-strand DNA (ssDNA) overhangs generated by 5′–3′ exonuclease resection of DSBs, similar to homologous recombination (HR)(Fig. 1b)[7,8]. The ability of Polθ to act on 3′ ssDNA overhangs to promote MMEJ of DSBs during replication explains how this specialized polymerase enables the proliferation of HR-deficient cells[5,8,9].

Polθ consists of a super-family 2 helicase (Polθ-hel), a disordered central domain (Polθ-cen), and an A-family polymerase (Polθ-pol) (Fig. 1a and Supplementary Fig. 1)[1,10,11]. Polθ-hel is related to HELQ/Hel308 helicases, which are involved in replication and repair[1,12,13]. Recent studies show that Polθ-hel unwinds short DNA in an ATP-dependent manner with 3′-5′ polarity, similar to HELQ/Hel308[12]. Polθ-hel also facilitates annealing of complementary ssDNA in an ATP-independent manner, and anneals RPA coated ssDNA by utilizing the energy of ATP[14]. Polθ-hel may also act as an anti-recombinase by counteracting RAD51 activity[15]. Polθ-pol is related to bacterial Pol I enzymes such as Klenow fragment, but contains an inactive 3′–5′ exonuclease domain[1,10–13,16], and exhibits low fidelity DNA synthesis and translesion (TLS) synthesis activities[17–21]. The polymerase and helicase include unstructured motifs, and a disordered motif (loop 2) in Polθ-pol was shown to promote its ssDNA extension and MMEJ activities[2,13,22].

The ssDNA overhangs that Polθ functions on during MMEJ have become clearer in recent cellular studies. Wyatt et al. demonstrated that relatively long (≥ 45–70 nt) ssDNA overhangs promote MMEJ in cells (Fig. 1b). This is in agreement with prior reports showing that MMEJ and HR share the same 5′–3′ DNA resection pathway involving MRN and CtIP[7], and that Polθ competes with HR[3,15]. Wyatt et al. also demonstrated that the efficiency of cellular MMEJ is positively correlated with 3′ ssDNA overhang length[8]. This report further demonstrated that Ku binds tightly to DNA ends with short 4 nt overhangs, but exhibits very low affinity for 70 nt overhangs[8]. This reveals a clear mechanistic difference between the respective substrate requirements for non-homologous end-joining (NHEJ) and MMEJ. How Polθ functions on long ssDNA overhangs to promote MMEJ, however, remains unknown.

## Results
**Polθ Promotes MMEJ of Long ssDNA.** Initial biochemical studies demonstrated that short (≤ 15 nt) 3′ overhangs are required for MMEJ activity by the isolated polymerase domain (Polθ-pol)[2]. Recent biochemical studies similarly show that Polθ-pol can perform MMEJ on short (< 12 nt) ssDNA[23]. Although these studies provide insight into the activity of Polθ-pol, they fail to recapitulate MMEJ on DNA with longer (45–70 nt) overhangs that support cellular MMEJ[8]. In vitro, Polθ-pol primarily performs snap-back replication on long ssDNA as a result of its intrastrand base-pairing activity[2,22]. For example, under physiological buffer conditions with $Mg^{2+}$, Polθ-pol efficiently extends relatively long (≥ 26 nt) ssDNA by using the 5′ portion of the ssDNA as a template *in cis* (Fig. 1c). Consistent with this, we demonstrate that Polθ-pol exclusively performs ssDNA extension (ssDNAx) when intrastrand base-pairing opportunities exist between the 3′ terminus and the 5′ portion of ssDNA (Supplementary Fig. 2). Hence, these and previous data demonstrate that the isolated polymerase strongly prefers intrastrand pairing on long ssDNA (Fig. 1c)[2,22]. Because endogenous Polθ facilitates MMEJ of DNA with long (≥ 45–70 nt) overhangs in cellular

studies[8], we hypothesized that the helicase domain within full-length Polθ (hereafter referred to as Polθ) promotes MMEJ by suppressing Polθ-pol intrastrand pairing. For instance, since Polθ-hel binds tightly to ssDNA and translocates along ssDNA with 3′–5′ directionality[12,13], it may inhibit Polθ-pol intrastrand pairing by binding and/or translocating along ssDNA upstream from the polymerase in the context of the full-length protein (Fig. 1d).

Technical difficulties in purifying recombinant human full-length Polθ has hindered our understanding of how this protein promotes MMEJ at the molecular level. Here, we purified recombinant human Polθ from *S. cerevisiae* using a N-terminal 3xFLAG-tag (Fig. 1e). Polθ and Polθ-pol exhibit identical primer extension activities (Fig. 1f), which is consistent with recent cellular studies indicating that Polθ-hel and Polθ-cen are dispensable for Polθ TLS activity on primer-templates[20]. Polθ exhibits ATPase activity as expected (Fig. 1g). We note that Polθ is stored in buffer with a high concentration of ATP. Thus, detection of Polθ hydrolysis of $^{32}P$-γ-ATP, which is present at substantially lower amounts than cold ATP, requires long incubation (Fig. 1g). Notably, Polθ remains active in ATP hydrolysis for several hours, demonstrating the enzyme is highly stable (Supplementary Fig. 3A).

Considering that Polθ acts on relatively long (≥ 45–70 nt) ssDNA overhangs to promote MMEJ in cells, we examined Polθ MMEJ of long ssDNA (70 nt and 100 nt) which models substrates with long 3′ overhangs (Fig. 1h). We find that Polθ performs efficient MMEJ of 70 and 100 nt ssDNA containing 3′ terminal 6 base pairs (bp) of microhomology (5′-CCCGGG-3′)(Fig. 1i, j). In contrast, Polθ-pol is deficient in MMEJ and predominantly performs ssDNAx on long ssDNA via snap-back replication (Fig. 1i, j), similar to prior studies and in Supplementary Fig. 2. Controls show that human Polθ-pol purified from different host organisms (*E. coli* and *S. cerevisiae*) are equally deficient in MMEJ of long ssDNA (Supplementary Fig. 3B). Endonuclease XmaI addition following MMEJ confirms the high molecular weight products are generated by end-joining (Fig. 1k, l). XmaI cleaves 5′-CCCGGG-3′ double-strand DNA. DNA sequencing also confirms Polθ MMEJ of long ssDNA (Supplementary Fig. 4A-4C). Remarkably, Polθ exhibits a > 10-fold higher efficiency of MMEJ on long ssDNA compared to Polθ-pol (Fig. 1m). MMEJ products accumulate slowly (≥ 2 min), indicating a slow rate-limiting interstrand pairing step (Fig. 1n). Polθ performs efficient MMEJ of long ssDNA even when added at a 30-fold lower concentration than ssDNA (Fig. 1o), and this activity is processive following initial extension of the minimally paired ends (Supplementary Fig. 3C). Polθ also performs MMEJ of long ssDNA containing non-palindromic 3′ terminal microhomology (Supplementary Fig. 3F, left).

As noted above, Polθ-pol promotes MMEJ of very short (≤ 15 nt) ssDNA which is almost entirely contained within its active site (Fig. 1p, left)(Supplementary Fig. 3D and 3E)[16,23]. On longer 26 nt ssDNA with identical microhomology, Polθ-pol exclusively promotes ssDNAx byproducts (Fig. 1p, right), which was confirmed by sequencing (Supplementary Fig. 4D-4F). Unexpectedly, full-length Polθ fails to extend short (≤ 26 nt) ssDNA (Fig. 1p), yet is active on a similar length primer-template (Fig. 1f). This suggests Polθ-cen and/or Polθ-hel restrict Polθ MMEJ activity to long ssDNA. This phenomenon is further investigated in Fig. 4. Importantly, the ability of Polθ-pol to perform MMEJ on short (≤ 12 nt) ssDNA (Fig. 1p, left; Supplementary Fig. 3D and 3E)[23], and short (≤ 15 nt) overhangs[2], demonstrates it performs interstrand pairing without Polθ-hel. Polθ-pol, however, fails to perform efficient MMEJ of long ssDNA under various conditions (Figs. 1i, j, q, and Supplementary Fig. 5). These data demonstrate that the helicase and/or

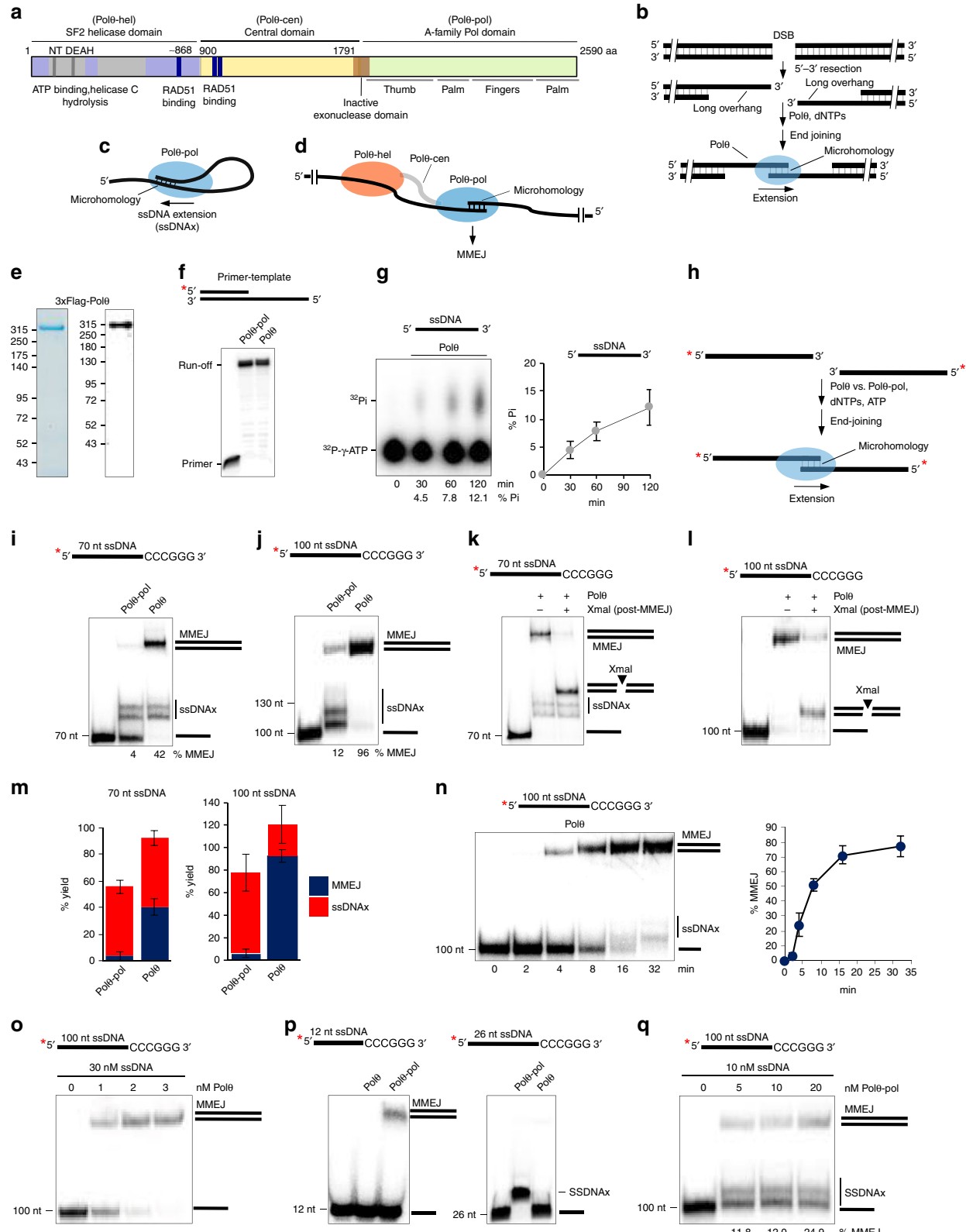

central domain promote Polθ MMEJ activity on long ssDNA while suppressing its activity on short ssDNA.

DNA length and sequence requirements for Polθ MMEJ were further explored. Consistent with Fig. 1, Polθ demonstrates superior MMEJ on the 70 and 100 nt ssDNA containing 6 bp microhomology compared to Polθ-pol (Fig. 2a, b). As mentioned above, cellular studies showed that MMEJ is positively correlated

with ssDNA overhang length[8]. Consistent with this, we find that decreasing ssDNA length to 45 nt while maintaining the same 6 bp microhomology significantly reduces Polθ MMEJ, yet, the full-length protein still shows significantly higher end-joining activity than Polθ-pol (Fig. 2c). Overall ssDNA length dependency for Polθ MMEJ on substrates with identical microhomology is plotted (Fig. 2d). As further controls for MMEJ, the addition of

**Fig. 1** Polθ promotes efficient MMEJ of long ssDNA. **a** Schematic of Polθ. **b** Schematic of MMEJ of DNA with long overhangs. **c** Schematic of Polθ-pol ssDNAx via snap-back replication. **d** Schematic of Polθ MMEJ of long ssDNA. **e** SDS gel (left) and Western blot (right) of purified human Polθ. **f** Denaturing gel showing primer extension by Polθ and Polθ-pol. **g** TLC plate showing ATP hydrolysis by Polθ (left). Plot of ATP hydrolysis in the presence of ssDNA. $n = 3 \pm$ s.d. (right). Initial rate of ATP hydrolysis = 738 pmol min$^{-1}$. **h** Schematic of MMEJ of long ssDNA. **i, j** Non-denaturing gels showing MMEJ of the indicated long ssDNA by Polθ and Polθ-pol at 3 nM. **k, l** Non-denaturing gels showing MMEJ by 3 nM of Polθ of the indicated ssDNA (lane 2) and subsequent cleavage by XmaI (lane 3). **m** Bar plots showing % of MMEJ and ssDNAx products generated by 3 nM Polθ and 3 nM Polθ-pol on the indicated ssDNA. $n = 3$ (left), $n = 4$ (right) $\pm$ s.d. **n** Non-denaturing gel showing time course of MMEJ by 3 nM of Polθ on the indicated ssDNA (left). Plot of Polθ MMEJ time course. $n = 3 \pm$ s.d. (right). Initial rate of MMEJ = 13.5 fmol MMEJ products min$^{-1}$. **o, q** Non-denaturing gels showing MMEJ by indicated amounts of Polθ (o) and Polθ-pol (q) on indicated concentrations of 100 nt ssDNA with 6 bp microhomology. **p** Non-denaturing gels of MMEJ reactions performed by 3 nM of the indicated proteins on short 26 nt and 12 nt ssDNA substrates with indicated microhomology. *, $^{32}$P. ssDNAx and MMEJ products are indicated. Source data are provided as a Source Data file

two different substrates (70 and 100 nt) with identical micro-homology yields the expected size MMEJ products (Fig. 2e). These data also confirm that Polθ exhibits preferential MMEJ on the longer 100 nt substrate when both oligos are present (Fig. 2e). Maintaining the same ssDNA length (70 nt) and microhomology tract length (6 bp), while reducing hydrogen bonds within the microhomology region has little effect on Polθ MMEJ efficiency (compare Fig. 2b and f). Polθ MMEJ efficiency is significantly reduced when microhomology length and hydrogen bonds are decreased to 4 bp and ≤ 12, respectively, on similar length (68–73 nt) ssDNA (data summarized in Fig. 2g; Fig. 2b, f, h–k shows gel data). Superior MMEJ activity by Polθ versus Polθ-pol on additional long ssDNA substrates is demonstrated (Supplementary Fig. 3F). Further controls show that preventing secondary ssDNA structures enables efficient Polθ MMEJ of long substrates (Supplementary Fig. 3I). Figures 1 and 2 reveal that Polθ is significantly more efficient in MMEJ of long ssDNA than Polθ-pol, and show that this activity is positively correlated with ssDNA length, microhomology length, and hydrogen bonding.

**Preventing Intrastrand Pairing Stimulates MMEJ by Polθ-Pol.** Our hypothesis predicts that the helicase stimulates Polθ-pol interstrand pairing by suppressing its dominant intrastrand pairing activity (Fig. 1d). The crystal structure of Polθ-pol demonstrates that only 8–10 bases of the primer strand are protected by the enzyme[16]. Assuming Polθ-pol remains in a similar conformation on ssDNA, the 5′ region of long ssDNA ( > 10 nt) would be extruded from the enzyme's DNA exit channel (Fig. 3a). Because a 5′ ssDNA extrusion greatly stimulates Polθ-pol ssDNAx via snap-back replication when intrastrand base-pairing is available (Fig. 3b and Supplementary Fig. 2), this prominent activity appears to outcompete interstrand pairing by the polymerase.

We predicted that preventing base-pairing opportunities between 3′ terminal bases and bases upstream along long the 5′ region of long ssDNA substrates would suppress intrastrand pairing and enable interstrand pairing by Polθ-pol (Fig. 3c). To test this, we examined Polθ-pol MMEJ activity on long ssDNA with complementary oligos pre-annealed to the 5′ region to inhibit intrastrand base-pairing. Consistent with the model in Fig. 3c, preventing intrastrand pairing by pre-annealing multiple complementary upstream oligos dramatically stimulates Polθ-pol MMEJ on a 100 nt substrate (compare Fig. 3g to Fig. 3d–f). In the scenario where no intrastrand pairing is possible, Polθ-pol MMEJ activity is identical to the full-length enzyme (Fig. 3g). Controls demonstrate that the high molecular product generated by Polθ and Polθ-pol on this long partial ssDNA (pssDNA) substrate is due to MMEJ as demonstrated by XmaI cleavage of its recognition site (double-stranded 5′-CCCGGG-3′; Fig. 3g). Further template controls using individual annealed complementary oligos containing unextendible 3′ terminal ends due the presence of 2′,3′-dideoxyribonucleotides fail to stimulate Polθ-pol

MMEJ (Fig. 3h–k). Hence, full inhibition of hydrogen bonding between 3′ and 5′ proximal bases is necessary for stimulating Polθ-pol MMEJ as predicted.

**Polθ-Helicase Attachment Facilitates MMEJ of Long ssDNA.** We next investigated how Polθ-hel contributes to MMEJ. Fluorescence anisotropy shows that Polθ exhibits a significantly lower $K_D$ than both Polθ-pol and Polθ-hel on short 29 nt fluorescein (FAM) conjugated ssDNA (Fig. 4a), demonstrating that Polθ-hel majorly contributes to Polθ ssDNA binding. Because Polθ-pol performs the interstrand pairing step as demonstrated by its MMEJ activity on short ssDNA (Fig. 1p, left; Supplementary Fig. 3D and 3E), and short overhangs (Fig. 3g)[2], and Polθ-hel strongly contributes to Polθ ssDNA binding, our data support a model whereby Polθ-hel binds ssDNA upstream from Polθ-pol (Fig. 4b). This protein-ssDNA conformation would ideally position Polθ-hel to block Polθ-pol intrastrand pairing in favor of interstrand pairing. Assays below further investigate this model.

To probe whether Polθ-hel ATPase function is necessary for stimulating MMEJ of long ssDNA, we purified PolθK121M which contains a methionine substitution for the conserved Walker A residue K121 that is essential for ATP binding and hydrolysis (Fig. 4c)[1,11–13,24]. PolθK121M is defective in ATPase activity as expected (Supplementary Fig. 3H). PolθK121M exhibits nearly identical primer extension activity compared to Polθ-pol (Fig. 4d), which is consistent with recent studies indicating that Polθ-hel does not contribute to Polθ TLS activity on primer templates in cells[20]. Remarkably, PolθK121M performs identical MMEJ compared to WT Polθ (Fig. 4e, f), demonstrating that Polθ ATPase activity is dispensable for MMEJ in our system. Consistent with this, WT Polθ performs efficient MMEJ in the presence of AMP-PNP, an ATP analog that cannot be hydrolyzed by Polθ-hel (Supplementary Fig. 3G)[12]. We note that ATP has no significant effect on Polθ-hel ssDNA binding (Supplementary Fig. 6E). Polθ ATPase activity may be involved in an auxiliary DNA repair function, such as DNA unwinding or dissociation of protein-ssDNA complexes[12,14,15]. The ATPase activity may also contribute to unresolved functions of Polθ such as in interstrand crosslink repair and mitochondrial DNA replication and repair[25,26].

To assess whether Polθ-cen contributes to Polθ MMEJ, we purified a Polθ mutant that includes a flexible 15 amino-acid (glycine-serine) linker in place of the central domain (PolθΔcen) (Fig. 4g, h)[27]. PolθΔcen exhibits normal primer extension activity (Fig. 4i), and is functional in ATP hydrolysis for over 3 h (Supplementary Fig. 6A). Remarkably, PolθΔcen shows identical MMEJ activity to WT Polθ (Compare Fig. 4j, k with Fig. 1i, n). Together, these data demonstrate that Polθ-hel alone is responsible for stimulating Polθ MMEJ of long ssDNA. The addition of purified Polθ-hel *in trans* has little effect on Polθ-pol MMEJ, even at relatively high concentrations (Fig. 4l). Controls show that Polθ-hel binds ssDNA (Fig. 4a, right), and exhibits

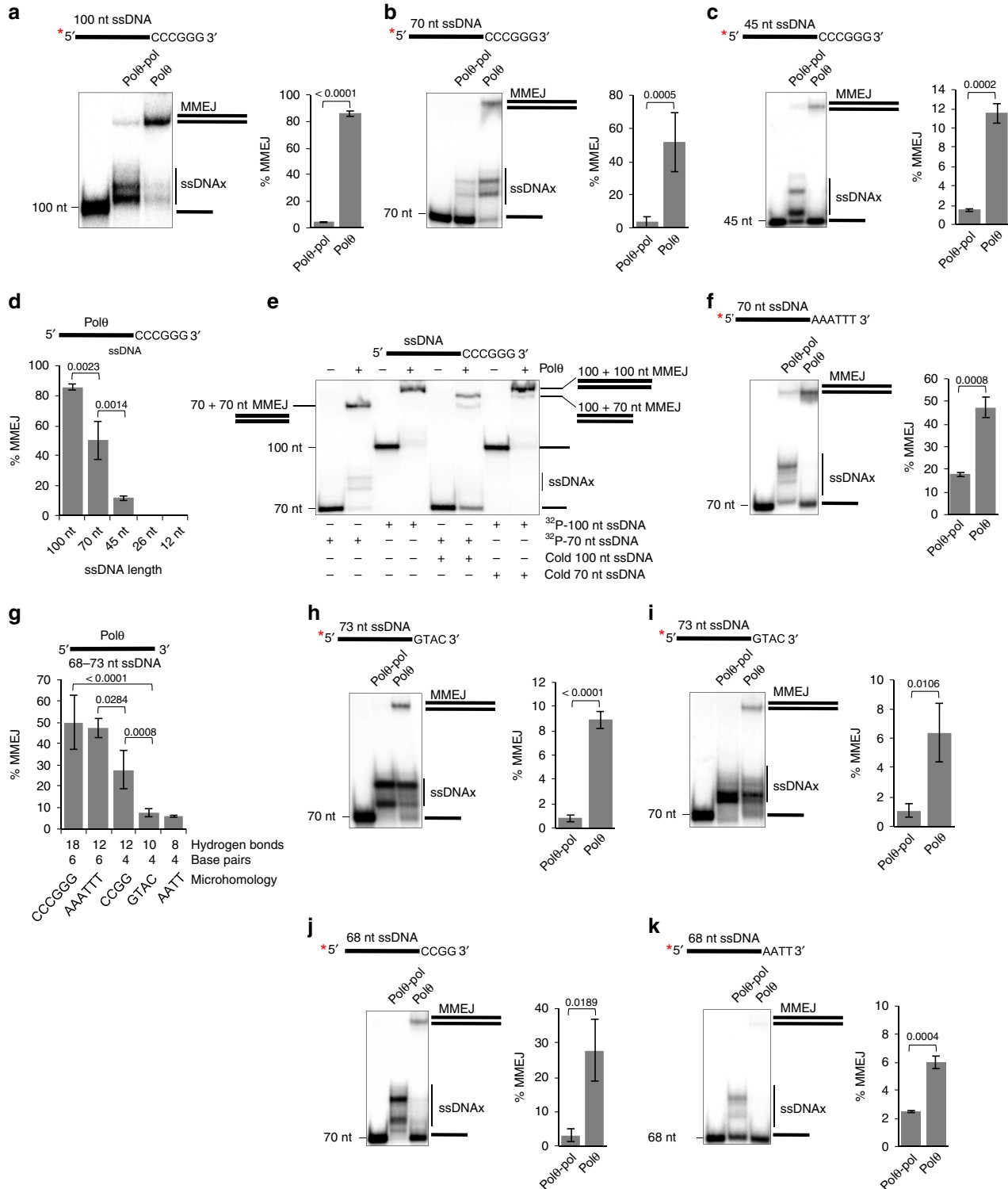

**Fig. 2** Substrate requirements for Polθ MMEJ of ssDNA. **a–c**, **f**, **h–k** Non-denaturing gels showing MMEJ reactions performed by 3 nM Polθ and 3 nM Polθ-pol on the indicated ssDNA substrates. 3′ terminal microhomology sequences are indicated (left). Bar plots showing % MMEJ (right). $n = 3 \pm$ s.d. **d** Bar plot showing % MMEJ by 3 nM Polθ on ssDNA of variable lengths with identical 6 bp microhomology. $n = 3 \pm$ s.d. **e** Non-denaturing gel showing MMEJ reactions performed by 3 nM Polθ in the presence of the indicated radio-labeled and cold ssDNA substrates. **g** Bar plot showing % MMEJ by 3 nM Polθ on 68–73 nt ssDNA substrates containing the indicated 3′ terminal microhomology sequences. $n = 3 \pm$ s.d. P values reported are from unpaired Student's two-sided t-test. *, [32]P. Source data are provided as a Source Data file

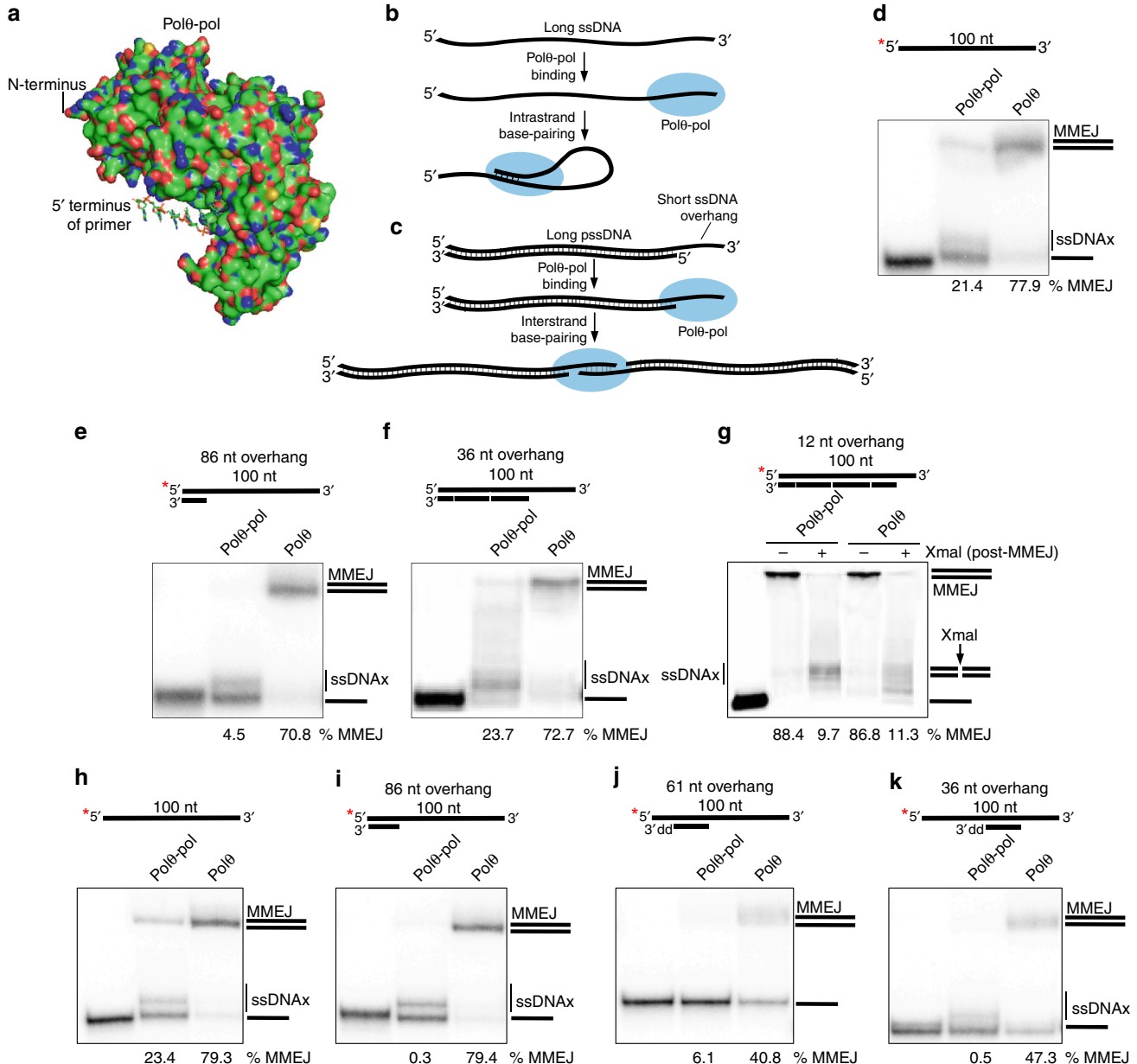

**Fig. 3** Preventing intrastrand base-pairing stimulates Polθ-pol MMEJ. **a** Structure of Polθ-pol containing the primer strand. PDB, 4 × 0Q[16]. **b**, **c** Schematic of Polθ-pol intrastrand (**b**) and interstrand (**c**) pairing. **d**–**k** Non-denaturing gels showing MMEJ reactions performed with 3 nM of indicated proteins and 10 nM of indicated DNA templates. Lanes 3 and 5 in **g** were treated with XmaI. % MMEJ is indicated. 3′dd, 2′3′-dideoxycytosine monophosphate. *, [32]P. Source data are provided as a Source Data file

ATPase activity (Supplementary Fig. 3A), and is therefore fully functional as in previous studies[12]. Taken together, our data demonstrate that Polθ-hel stimulates Polθ-pol MMEJ *in cis*, by enabling interstrand pairing by the polymerase domain, and show that the mechanistic function of Polθ-cen is to tether Polθ-hel to Polθ-pol (Fig. 4b).

**Central Domain Regulates Polθ MMEJ Substrate Requirements.** Because Polθ fails to perform MMEJ or ssDNAx on short (≤ 26 nt) ssDNA compared to Polθ-pol (Fig. 1p), we tested whether Polθ-cen was responsible for this autoinhibitory function. Indeed, PolθΔcen exhibits both MMEJ and ssDNAx activities on 26 nt ssDNA containing 6 bp microhomology, whereas WT Polθ and PolθK121M which contain Polθ-cen show no activity on this substrate (Fig. 4m; Supplementary Fig. 6B). Polθ-

pol exclusively performs ssDNAx on this substrate as in previous studies (Fig. 1p, right; Fig. 4n; Supplementary Fig. 4D-4F)[2], and the addition of Polθ-hel *in trans* fails to stimulate Polθ-pol MMEJ of this substrate as expected (Supplementary Fig. 6C). PolθΔcen also performs MMEJ of short 12 nt ssDNA, like Polθ-pol (Supplementary Fig. 3D). Although WT Polθ and PolθK121M fail to extend short ssDNA due to the presence of Polθ-cen (Fig. 4m), they exhibit efficient activity on short primer-templates (Figs. 1f and 4d). Thus, our data reveal a specific regulatory function for Polθ-cen in restricting Polθ MMEJ activity to long ssDNA which models resected DSBs.

**Polθ Cooperativity Underlies MMEJ.** Because covalent attachment of Polθ-hel to Polθ-pol is the minimal requirement for stimulating Polθ-pol MMEJ on relatively long ssDNA substrates,

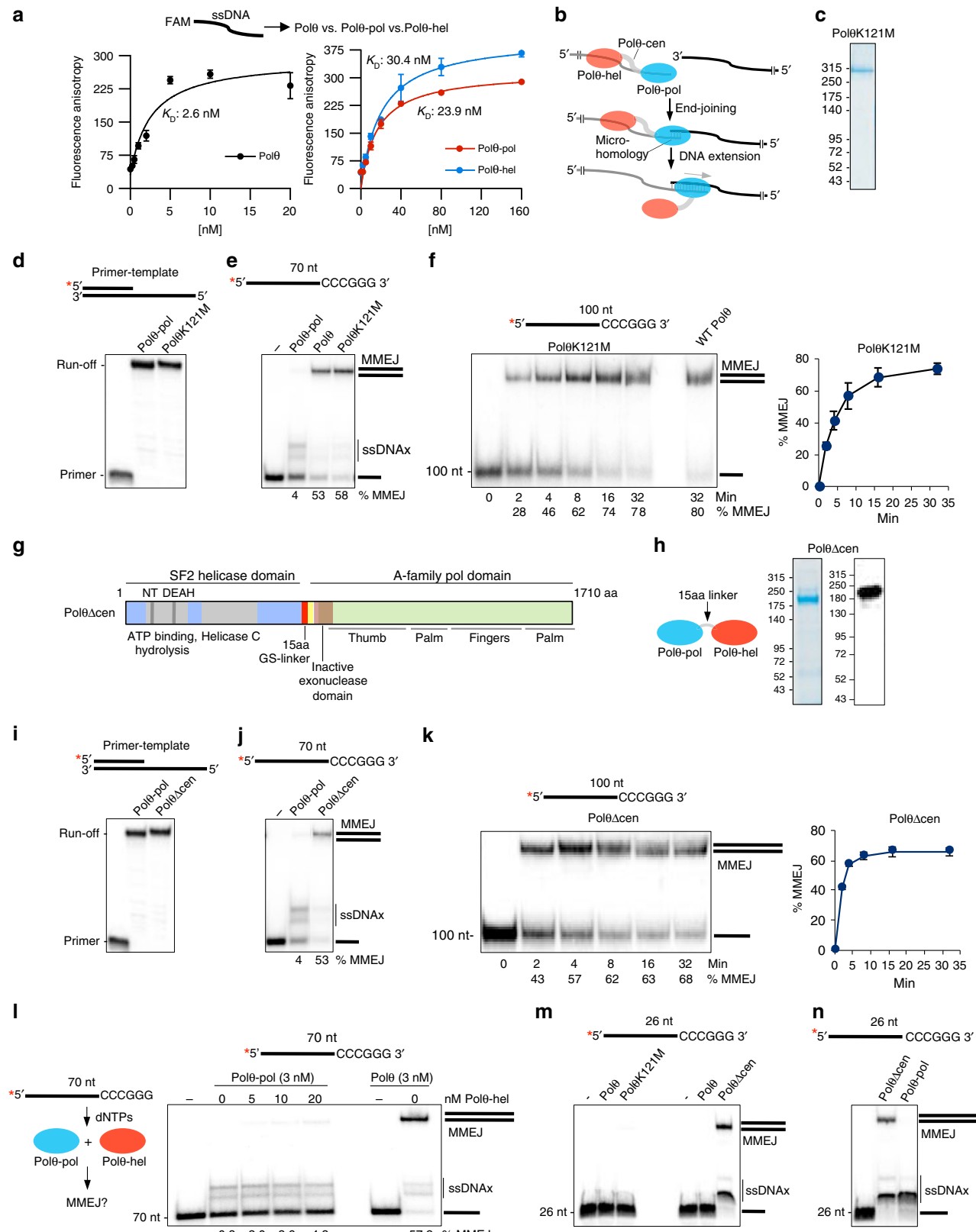

this indicates close cooperation between these two domains on long ssDNA. To gain insight into how these linked enzymes cooperate, we modeled the structure of the minimal end-joining PolθΔcen protein on long ssDNA which combines the crystal structures of Polθ-pol and Polθ-hel attached by a flexible linker (Fig. 5a)[13,16]. Our previous studies demonstrate that Polθ-hel

exhibits similar activities to the related helicase Hel308, such as ATP-dependent 3′-5′ directional ssDNA translocation activity[12]. Based on this polarity, Polθ-hel was modeled facing towards the 5′ terminus. The model superimposes the ssDNA substrate from the Hel308 structure, in which the enzyme translocates along, onto Polθ-hel[28]. The primer strand from the Polθ-pol structure is

**Fig. 4** Polymerase-helicase attachment stimulates MMEJ of long ssDNA. **a** Plots of fluorescence anisotropy in the presence of the indicated amounts of Polθ (left), and Polθ-pol and Polθ-hel (right). n = 3 ± s.d. $K_D$ values indicated. **b** Model of Polθ MMEJ of long ssDNA. **c** SDS gel of PolθK121M protein. **d, i** Denaturing gels showing primer extension by 5 nM of indicated proteins. **e, j** Non-denaturing gels showing MMEJ reactions performed with 3 nM of the indicated proteins on the indicated ssDNA. **f, k** Non-denaturing gel showing a time course of MMEJ of the indicated ssDNA by 3 nM of the indicated proteins (left). Plot of time course products. n = 3 ± s.d. (right). Initial rate of MMEJ = 20.7 fmol MMEJ products min$^{-1}$ **f**, 42 fmol MMEJ products min$^{-1}$. **k, g** Schematic of PolθΔcen construct. **h** Cartoon of PolθΔcen (left). SDS gel (middle) and Western blot (right) of purified PolθΔcen. **l** Schematic of MMEJ assay with Polθ-hel and Polθ-pol added *in trans* (left). Non-denaturing gel showing MMEJ reactions performed by 3 nM Polθ-pol or Polθ with and without the indicated amounts of Polθ-hel. **m, n** Non-denaturing gel showing MMEJ reactions performed by 3 nM of the indicated proteins on the indicated ssDNA. % MMEJ indicated. *, $^{32}$P. Source data are provided as a Source Data file

also included with additional ssDNA linking the two substrates together, resulting in a long contiguous 31 nt ssDNA that is easily bound by the tethered enzymes. In this close configuration along ssDNA Polθ-pol and Polθ-hel fit well together and space is readily available for a long flexible linker (e.g. Polθ-cen). Although this model is speculative, it suggests bivalent ssDNA binding by the fusion protein which would be expected to be in the low nanomolar range.

To directly probe whether Polθ-hel binds upstream from Polθ-pol within the context of the minimal PolθΔcen end-joining protein, we used nuclease protection assays. First, we probed PolθΔcen versus Polθ-pol protection of a 3′ Cy3 conjugated 34 nt ssDNA using bacteriophage T5 5′-3′ exonuclease. Polθ-hel binding to the 5′ region of ssDNA is expected to block 5′-3′ exonuclease activity. In support of this, the presence of PolθΔcen protects most of the ssDNA from 5′ to 3′ exonuclease activity as indicated by the prominent bands at 34 nt and 26 nt (Fig. 5b, lane 4). Although the exonuclease does not fully digest the naked DNA under these conditions (lane 2), only the higher molecular weight (26 nt and 34 nt) bands are observed when PolθΔcen is bound (lane 4). A five-fold higher concentration of T5 exonuclease similarly reveals a PolθΔcen footprint of at least 26 nt (Supplementary Fig. 6D). Because Polθ-hel binds to both ssDNA and double-strand DNA[13], we further investigated the footprint of PolθΔcen on partial ssDNA (pssDNA) of similar size (Fig. 5c). The pssDNA substrate contains an EcoRI site within the 5′ double-stranded portion. Polθ-pol or PolθΔcen was incubated with the pssDNA for 15 min followed by the addition of EcoRI. The results show that PolθΔcen blocks EcoRI cleavage whereas Polθ-pol does not (Fig. 5c), which further supports Polθ-hel binding upstream from Polθ-pol (Fig. 5a).

The model in Fig. 5a supports the idea that Polθ-hel promotes Polθ-pol interstrand pairing by preventing its intrastrand pairing activity. To further investigate this mechanism, we probed whether polymerase-helicase attachment stimulates interstrand pairing by utilizing fluorescence resonance energy transfer (FRET)(Fig. 5d). PolθΔcen produces a significant increase in FRET in a concentration dependent manner, whereas Polθ-pol and Polθ-hel fail to produce significant FRET above background (Fig. 5d). This demonstrates that polymerase-helicase attachment is necessary for promoting stable DNA synapsis which is consistent with helicase stimulation of Polθ-pol interstrand pairing. The nuclease protection and FRET assays in Fig. 5b–d is supported by the PolθΔcen:ssDNA model in Fig. 5a. Although we were unable to concentrate WT Polθ to sufficient levels for the nuclease and FRET assays, we reiterate that PolθΔcen performs MMEJ in an identical manner to WT Polθ on long ssDNA substrates and is even permissive for MMEJ on short ssDNA substrates, and therefore serves as a valid model enzyme for probing mechanisms of MMEJ.

The data in Fig. 5d demonstrate that at least a one to one ratio of PolθΔcen (20 nM) to ssDNA (20 nM) is needed for saturation of FRET. This suggests that at least two molecules of PolθΔcen

are involved in the DNA synapsis step, which supports a dimeric model of end-joining (Fig. 5e). Consistent with this, prior structural studies have revealed a dimeric model of polymerase-dependent MMEJ. For example, the polymerase domain of prokaryotic NHEJ factor LigD (LigD-pol) was shown to facilitate microhomology-mediated DNA synapsis by forming a head-to-head dimer conformation at the 3′ ssDNA termini[29]. Interestingly, both Polθ-pol and LigD-pol utilize a solvent exposed loop near their DNA entrance channel to facilitate MMEJ, which suggests a similar mechanism of DNA synapsis/MMEJ by these multi-functional end-joining polymerases[2,29]. We reiterate that Polθ-pol is efficient in MMEJ on short ssDNA and short overhangs, and thus can clearly perform MMEJ in the absence of Polθ-hel. This unequivocally places Polθ-pol at the 3′ terminus of ssDNA like LigD-pol.

To further probe the dimeric model of DNA synapsis/MMEJ (Fig. 5e), we took advantage of the mechanistic requirement for WT Polθ end-joining activity on long ssDNA. Because Polθ only performs efficient MMEJ on long (≥ 70 nt) ssDNA, this indicates that the full-length protein becomes activated for end-joining on long substrates, potentially due to structural reconfiguration of Polθ-cen which specifically suppresses Polθ activity on short ssDNA. Regardless of the mechanism by which Polθ becomes activated on long ssDNA, if two long substrates are required for Polθ MMEJ, this would indicate that at least two activated molecules of the full-length protein are necessary for facilitating MMEJ (Fig. 5f, left). On the other hand, if Polθ can efficiently join a long substrate to a short substrate, this would demonstrate that only one activated molecule of Polθ is needed for MMEJ on the long substrate and support a monomeric mechanism of end-joining (Fig. 5f, right). To test these models, we examined whether Polθ can efficiently join a long 70 nt ssDNA to a short 26 nt ssDNA. Consistent with our results from above, controls show that Polθ performs MMEJ on 70 nt ssDNA but not 26 nt (Fig. 5g, lanes 1–4). The addition of equimolar amounts of cold 70 nt ssDNA and radiolabeled 26 nt ssDNA shows no end-joining between these substrates (lanes 5–6). Likewise, Polθ shows little or no end-joining between the 70 nt and 26 nt substrates when either the 70mer is radiolabeled (lanes 7–8) or both substrates are labeled (lanes 9–10). Hence, these data which are summarized at the right of Fig. 5g indicate that at least two active molecules of Polθ—one on each long strand—are needed for facilitating efficient MMEJ which supports the dimeric model of MMEJ (Fig. 5f, left). Together, the data in Fig. 5 highlight the importance of cooperativity between Polθ molecules and subdomains in MMEJ.

**Polθ Forms Complexes that Promote DNA Accumulation and MMEJ.** Because our data indicate that at least two Polθ molecules participate in DNA synapsis/MMEJ, we probed the oligomeric state of the full-length protein. X-ray crystallography demonstrates that Polθ-hel and Polθ-pol can form homo- tetramers and

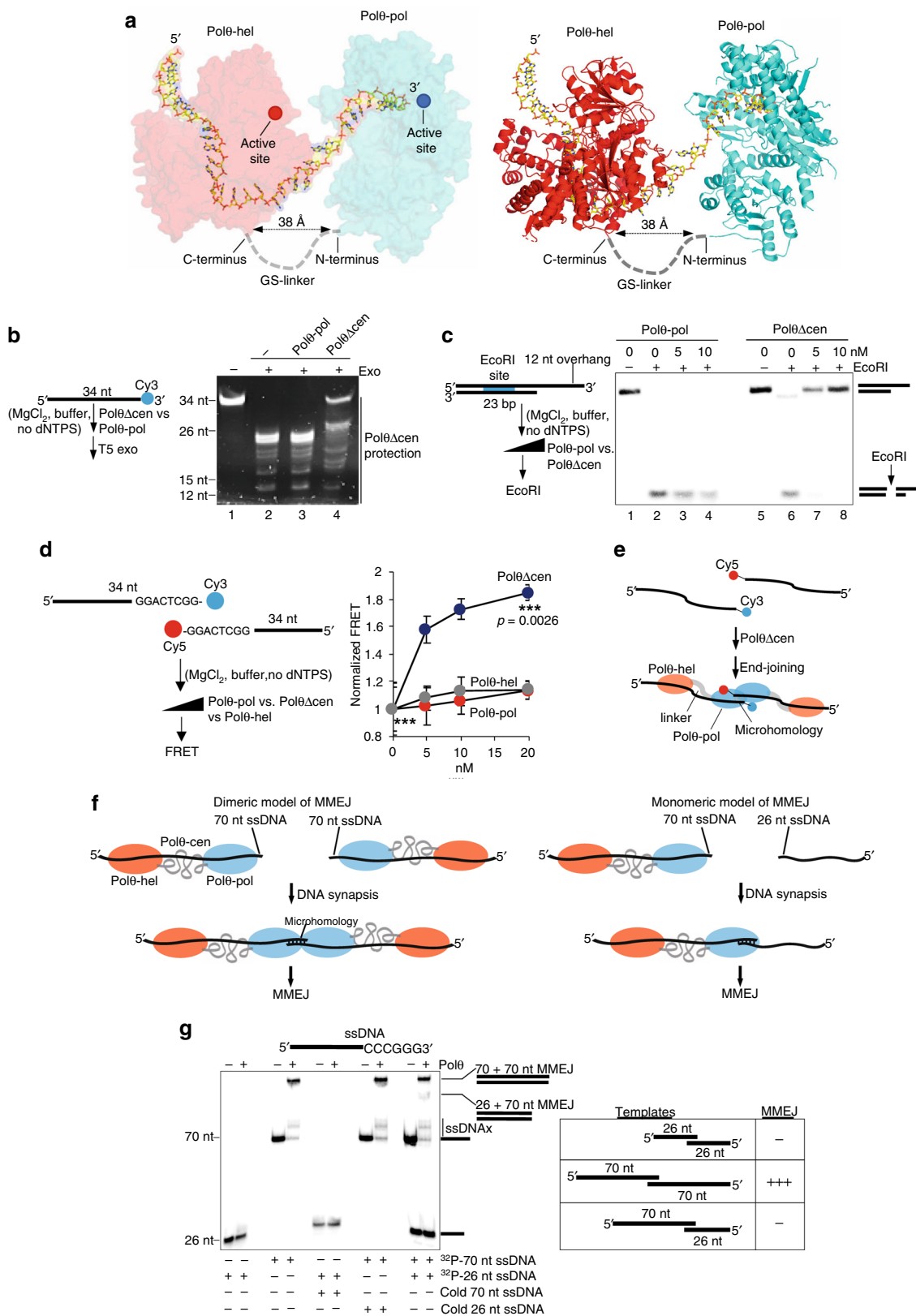

dimers, respectively, which suggests that the full-length protein may also form multimers[13,16]. The oligomerization state of WT Polθ with and without 26 nt ssDNA was investigated using negative staining electron microscopy (EM) following incubation in MMEJ buffer conditions lacking dNTPs and BSA. Without ssDNA, 81% of Polθ behaves as monomers and dimers, whereas

3% correspond to large (>1000 Å) multimers (Fig. 6a, b). Pre-incubation with ssDNA caused a 9-fold increase in large >1000 Å Polθ multimers, and corresponding decreases in monomers and dimers (Fig. 6a, b). ssDNA-dependent Polθ multimerization was also observed via scanning force microscopy (SFM) using the same buffer conditions (Fig. 6e, f). Here, 62% of Polθ appears as

**Fig. 5** Cooperation of Polθ molecules and Polθ subdomains promotes ssDNA synapsis and MMEJ of long ssDNA. **a** Structural model of PolθΔcen on long ssDNA. PDB 4×0Q (Polθ-pol, blue), 5AGA (Polθ-hel, red). Surface (left) and ribbon (right) representations shown. **b** Exonuclease footprinting of PolθΔcen on ssDNA. Schematic of assay (left). Denaturing gel showing T5 exonuclease (0.1 U/ul) digestion of the indicated ssDNA with 10 nM Polθ-pol (lane 3), 10 nM PolθΔcen (lane 4), or no protein (lane 2). **c** EcoRI probing of PolθΔcen pssDNA binding. Schematic of assay (left). Non-denaturing gel showing EcoRI cleavage of the indicated pssDNA in the presence of the indicated proteins. **d** Schematic of FRET assay (left). Plot showing normalized FRET by the indicated proteins. $n = 3 \pm$ s.d. $P$ values, unpaired Student's two-sided $t$-test. **e** Dimeric model of PolθΔcen MMEJ of ssDNA with fluorophores. **f** Models of MMEJ involving Polθ monomers and dimers. **g** Polθ requires two long ssDNA substrates for MMEJ. Non-denaturing gel showing MMEJ by 3 nM of Polθ in the presence of the indicated ssDNA substrates (left). Table summarizing MMEJ data (right). Source data are provided as a Source Data file

tetramers or larger with ssDNA (Fig. 6e, f). Interestingly, EM and SFM reveal that PolθΔcen forms large (100–500 nm) multimers even in the absence of ssDNA (Fig. 6c, d, g). This indicates that Polθ-cen suppresses Polθ multimerization when DNA is absent. Further imaging provides evidence that Polθ-hel ssDNA binding primarily contributes to ssDNA-dependent Polθ multimerization (Supplementary Fig. 7A-7C).

Polθ multimers can conceivably facilitate DNA synapsis by inducing high local DNA concentration. To test for Polθ-mediated ssDNA accumulation, WT and Polθ mutants were mixed with Cy3-conjugated ssDNA 46 nt in length in MMEJ buffer conditions without dNTPs, then Cy3 was imaged using confocal microscopy. Polθ, PolθK121M and PolθΔcen promote an increase in local Cy3-ssDNA concentration indicated by high intensity Cy3-ssDNA complexes (Fig. 6h, i; Supplementary Movies 1 and 2). In contrast, Polθ-pol and Polθ-hel do not promote Cy3-ssDNA complexes (Fig. 6h, i). The frequency of Cy3-ssDNA complexes was significantly higher with PolθΔcen (Fig. 6h, i), which is consistent with the inherent multimeric state of PolθΔcen (Fig. 6c, d, g). PolθΔcen complexes increase in size over time, likely due to complex fusion (Fig. 6j, k; Supplementary Movie 3). We note that the EM, SFM and confocal microscopy assays include 30 mM NaCl, 0.01% NP-40 and 2 mM ATP which acts as a hydrotrope to promote protein solubility[30]. Thus, the observed complexes are not likely due to protein aggregation or precipitation. Consistent with this, further controls below provide direct evidence that these large multimeric complexes are fully active in MMEJ (Fig. 6l–n). As another control, we show that RAD52, which forms large multimers and performs a related form of ssDNA end-joining involving microhomology, promotes similar Cy3-ssDNA complexes (Supplementary Fig. 7D). The data in Fig. 6h, i demonstrate that polymerase-helicase tethering facilitates ssDNA accumulation by enabling a particular form of Polθ oligomerization.

We reiterate that PolθΔcen is active in MMEJ on both long and short ssDNA due to lack of Polθ-cen autoinhibitory activity, and therefore serves as a model end-joining protein on various substrates. Since PolθΔcen readily forms large dynamic complexes with ssDNA (Supplementary Movie 3), we examined whether these complexes are active by imaging PolθΔcen MMEJ of Cy3-ssDNA with 6 bp 3′ terminal microhomology in real-time (Fig. 6l–n; Supplementary Movie 4). Here, Cy3-ssDNA both prior to and following the reaction was resolved in a non-denaturing gel after proteinase K treatment, identical to other reactions (Fig. 6l, n). The observed multimeric PolθΔcen-Cy3-ssDNA complexes convert nearly all of the Cy3-ssDNA into MMEJ products, demonstrating that Polθ-ssDNA multimeric complexes are active in end-joining (Fig. 6l–n). Both WT Polθ and PolθΔcen multimeric complexes increase in total area during MMEJ, suggesting a gel-like phase separation of DNA during end-joining (Supplementary Fig. 7E and 7F). SFM demonstrates that large Polθ multimers also promote the accumulation of long (1.8 kb) double-strand DNA (Fig. 6o; Supplementary Fig. 7G-7K), and this phenomenon occurs with both 30 mM and 100 mM NaCl

(Supplementary Fig. 7L). Taken together, these imaging data demonstrate that Polθ forms multimeric complexes on DNA which facilitate DNA accumulation and end-joining, and support the notion that at least two molecules of Polθ cooperate in MMEJ.

To gain further insight into the molecular architecture of Polθ multimeric complexes, we performed molecular dynamics simulations of systems containing thousands of monomers of the minimal PolθΔcen end-joining protein. To overcome length- and time- scale limitations intrinsic to any atomistic description of complex molecular systems, we resorted to a coarse-grained representation of Polθ that retained the major structural features of this protein. For example, we simulated a minimalist model of PolθΔcen consisting of a combination of two hard-spheres (Polθ-pol and Polθ-hel) connected by a flexible linker (Fig. 7a). The sphere radii were chosen to match the Polθ-hel and Polθ-pol X-ray crystallography structures, and the linker length corresponds to its sequence[13,16]. Importantly, we assumed attractive interactions between pairs of hard spheres of the same kind, since Polθ-pol and Polθ-hel form homo- dimers and tetramers, respectively (Fig. 7b). Despite these simple rules, the simulated system showed an intriguing phase behavior: the monomers quickly assembled into complexes of diverse size and shape (Supplementary Movies 5–8).

Overall, there is a close match between the morphology of the simulated complexes and the EM images (Fig. 7c, d). A peculiar feature of these complexes is their density: despite the fact that the average number of nearest neighbors surrounding each monomer is similar to the close packing value, the volume occupied by proteins in each complex is only 5%. Another important feature is the shape of the complexes: in liquid-liquid phase-separation processes droplets assume a spherical shape in order to minimize surface area[31]. The highly irregular shape of the assemblies suggests that, rather than droplets, the proteins form extended percolating networks; moreover, the fact that they are catalytically active indicates that the assemblies are permeable to the solvent and, possibly, large solutes. Based on the computational and EM evidence, we surmise that Polθ, and especially PolθΔcen, form large (>100 nm) gel-like complexes similar to others reported[32,33], which accumulate DNA due to the high local concentration and accessibility of the polymerase and helicase domains.

## Discussion

Using a structure function approach in vitro, we investigated how recombinant human full-length Polθ promotes MMEJ. On short ssDNA and short 3′ overhangs, Polθ-pol efficiently performs MMEJ due to its interstrand pairing activity (Fig. 7i). Thus, Polθ-pol is capable of MMEJ without the Polθ-hel on short ssDNA substrates, and this places the polymerase at the 3′ terminus of ssDNA during end-joining. On longer ssDNA substrates, Polθ-pol is unable to perform efficient MMEJ, and instead almost exclusively performs snap-back replication due to intrastrand base-pairing (Fig. 7f). To reconcile the inability

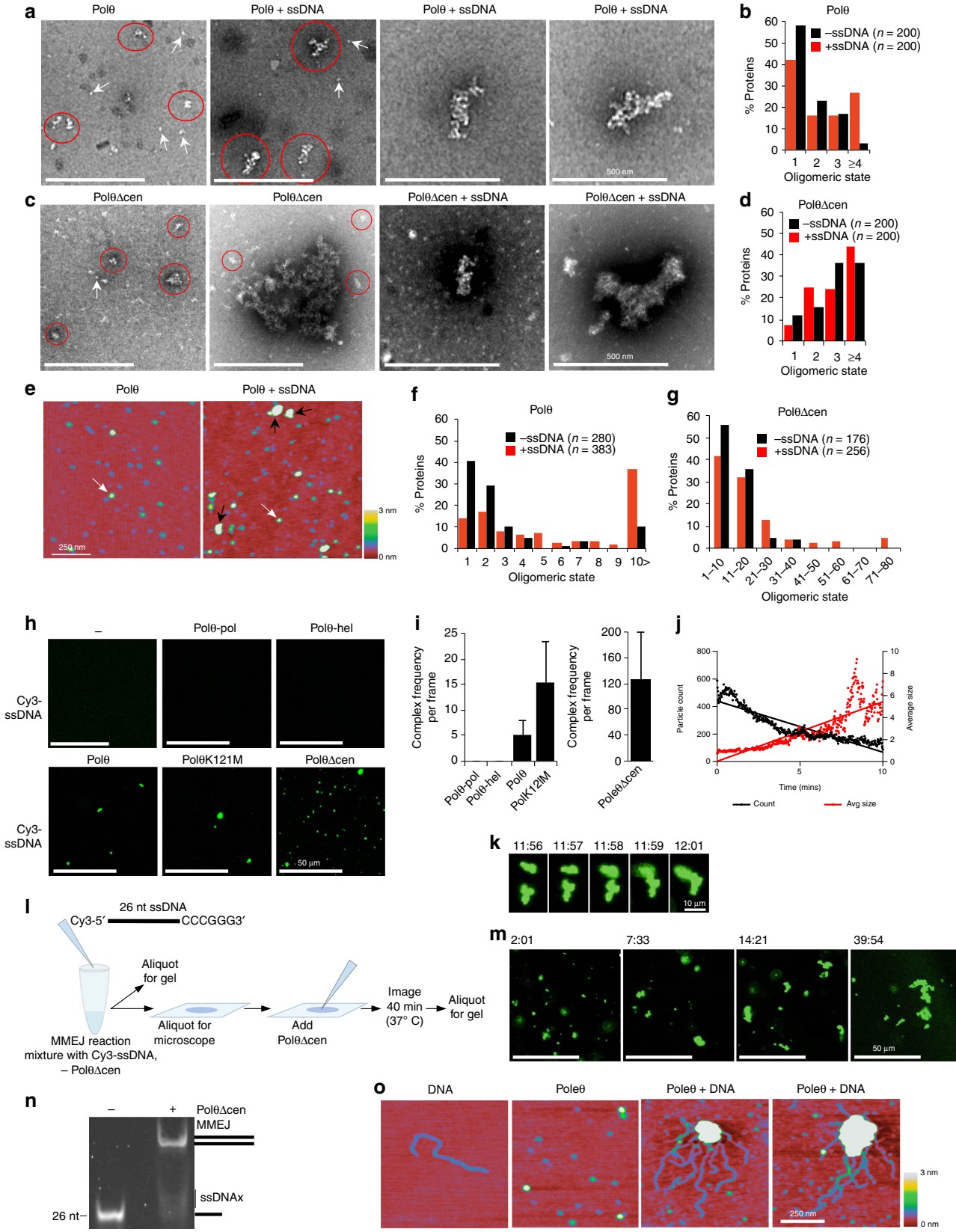

of Polθ-pol to join together long ssDNA overhangs which facilitate MMEJ in cells, we hypothesized that the full-length protein is necessary for this activity. Indeed, Polθ-hel dramatically upregulates Polθ-pol MMEJ of long ssDNA (Fig. 7e), by suppressing unproductive Polθ-pol snap-back replication

(Fig. 7f). Polθ-hel stimulation of Polθ-pol end-joining requires its attachment to the polymerase, but does not depend on Polθ-cen (Fig. 7g). Thus, Polθ-cen acts as a flexible linker, but also regulates the structure and function of Polθ as discussed below.

**Fig. 6 Polθ forms multimeric complexes that promote DNA accumulation and MMEJ. a, c** Negative staining EM showing single particles of Polθ (**a**) and PolθΔcen (**c**) with and without 26 nt ssDNA. White arrows indicate protein monomers. Red circles indicate protein multimers. Scale bar = 500 nm. **b, d** Bar chart showing oligomeric states of Polθ (**b**) and PolθΔcen (**d**) with (red) and without (black) 26 nt ssDNA determined by EM. **e** SFM images of Polθ with (right) and without (left) 26 nt ssDNA. White arrow indicates Polθ monomer. Black arrow indicates Polθ multimers. Blue particles represent small buffer components. Scale bar = 250 nm. **f, g** Bar charts showing oligomeric states of indicated proteins with (red) and without (black) 26 nt ssDNA determined by SFM volume measurements. **h** Confocal microscopy images of 46 nt Cy3-ssDNA alone (upper left) and with the indicated proteins. Scale bar = 50 μm. **i** Bar chart showing frequency of 46 nt Cy3-ssDNA complexes per frame with the indicated proteins. n = 3 ± s.d. **j** Scatter plot showing average size and frequency of PolθΔcen-Cy3-ssDNA particles over time. **k** Time lapse confocal microscopy images showing fusion of multimeric PolθΔcen-Cy3-ssDNA particles. ssDNA length, 46 nt. Imaging times are indicated. **l** Schematic of MMEJ reaction performed under the microscope and imaged in real-time. **m** Time lapse confocal microscopy images of PolθΔcen-Cy3-ssDNA complexes formed during MMEJ reaction. ssDNA length, 26 nt. Image times are indicated. **n** Non-denaturing gel showing efficient MMEJ of the 26 nt Cy3-ssDNA containing 6 bp microhomology performed by PolθΔcen during confocal microscopy imaging. **o** SFM images of DNA (left), Polθ (second panel), and Polθ-DNA complexes (right panels). DNA length, 1.8 kb. Scale bar = 250 nm. Source data are provided as a Source Data file

Polθ-hel stimulates Polθ-pol end-joining independently from its ATPase function. This supports the notion that Polθ-hel upregulates Polθ-pol MMEJ on long overhangs by simply binding ssDNA 5′ proximal to the polymerase where it suppresses unproductive intrastrand pairing and snap-back replication by Polθ-pol (Fig. 7e, g). Polθ-hel stimulation of Polθ-pol MMEJ is directly correlated to ssDNA length, which further supports a configuration where Polθ-hel binds upstream from Polθ-pol (Fig. 7e, g). For instance, helicase tethering to the polymerase shows no stimulation of Polθ-pol MMEJ on short substrates where snap-back replication is not possible (Fig. 7j). On the other hand, polymerase-helicase tethering dramatically stimulates Polθ-pol MMEJ on long substrates (Fig. 7e, g). Thus, polymerase-helicase attachment exclusively provides a mechanistic advantage on long ssDNA where Polθ-hel can bind upstream.

Polθ is unable to perform MMEJ or ssDNAx on short (≤ 26 nt) ssDNA (Fig. 7h). However, it is active on short primer-templates. In contrast, PolθΔcen is fully functional on short ssDNA (Fig. 7j), and behaves identically to WT Polθ on long ssDNA (Fig. 7g) and primer-templates. These data reveal an autoinhibitory function for Polθ-cen on short ssDNA (Fig. 7h), which may regulate Polθ substrate selection in cells. Because Polθ is activated for MMEJ on longer ssDNA, the autoinhibitory function of Polθ-cen becomes suppressed, presumably due to its conformational change upon Polθ binding to long ssDNA (Fig. 7e). Intriguingly, two long ssDNA substrates are essential for efficient MMEJ by Polθ, which indicates that two activated Polθ molecules, for example one on each strand, participate in end-joining. Based on this, we propose that head-to-head Polθ complexes facilitate the initial DNA synapsis step of MMEJ, similar to prokaryotic NHEJ factor LigD-pol (Fig. 7e)[29,34].

Polθ forms multimeric complexes upon binding DNA. Replacement of Polθ-cen with a short linker, however, results in multimerization of the enzyme into an extended gel-like phase, even without DNA. This suggests that Polθ-cen suppresses Polθ oligomerization by masking protein-protein interactions. Polθ DNA binding may alter the conformation of Polθ-cen which in turn enables Polθ oligomerization via protein-protein interactions. Our imaging data demonstrate that Polθ multimers facilitate DNA accumulation and MMEJ. In summary, this report reveals the importance of the unique polymerase-helicase architecture of Polθ which is essential for MMEJ.

## Methods

**MMEJ**. 10 nM 5′-[32]P radiolabeled ssDNA in buffer (25 mM Tris-HCl, pH 8.8, 1 mM DTT, 0.01% NP-40, 0.1 mg/ml BSA, 10% glycerol, 10 mM MgCl₂, 2 mM ATP, 20 μM dNTPs, 30 mM NaCl); the reaction was initiated by the addition of the indicated Polθ enzyme and was incubated for 45 min or as indicated at 37 °C. For analysis in nondenaturing gels, reactions were terminated by the addition of non-denaturing stop buffer (100 mM Tris–HCl, pH 7.5, 10 mg/ml proteinase K, 80 mM EDTA, and 0.5% SDS) and incubated at 37 °C for at least 15–20 min. DNA was resolved in non-denaturing 11% or 12% polyacrylamide gels and analyzed by phosphorimager (Fujifilm FLA 7000). For time course experiments an aliquot of sample was removed from pooled reactions at the specified time point and transferred to tubes containing non-denaturing stop buffer. All quantified experiments were performed in triplicate and plotted with ± s.d. Quantification was performed using ImageJ Gel Analysis. For XmaI digestion assays, after initial incubation with the indicated Polθ enzyme the following buffer was added (50 mM Potassium Acetate, 20 mM Tris-acetate, pH 7.9, 10 mM Magnesium Acetate, 100 μg/ml BSA). 25 units of XmaI (New England Biolabs) was then added, as indicated, and incubated overnight at 37 °C. The reaction was then stopped using non-denaturing stop buffer and resolved as above.

**Primer extension**. 10 nM 5′-[32]P radiolabeled pssDNA in buffer (25 mM Tris–HCl, pH 8.8, 1 mM DTT, 0.01% NP-40, 0.1 mg/ml BSA, 10% glycerol, 10 mM MgCl₂, 2 mM ATP, 20 μM dNTPs, 30 mM NaCl); the reaction was initiated by the addition of the indicated Polθ enzyme and incubated for 30 min at 37 °C. For analysis in denaturing gels, reactions were terminated by the addition of denaturing stop buffer (90% formamide and 50 mM EDTA). DNA was resolved in denaturing 15% polyacrylamide gels and analyzed by phosphorimager (Fujifilm FLA 7000).

**ATPase assay**. The indicated amounts of proteins were incubated with 10 μM ATP, 2 μCi of (γ-[32]P) ATP and 100 nM ssDNA (29 nt poly-dT) in buffer (50 mM Tris-HCl, pH 7.5, 10 mM MgCl₂, 5 mM DTT, 0.1 mg/ml BSA, and 10% v/v glycerol) at room temp for the indicated times. The reaction mixture was then spotted onto a TLC plate on PEI cellulose, which was developed in a buffer containing 1 M acetic acid and 0.25 M LiCl₂ for 1.5 h. Plates were dried, then visualized by phosphorimager (Fujifilm FLA 7000).

**Nuclease protection assays**. T5 exonuclease assay: 8 nM 3′-Cy3 ssDNA (RP540Cy3) in buffer (50 mM Potassium Acetate, 20 mM Tris-acetate, pH 7.9, 10 mM Magnesium Acetate, 1 mM DTT, 30 mM NaCl) was mixed with 10 nM PolθΔcen or Polθ-pol and was preincubated for 10 min at 37 °C. After initial incubation, T5 Exonuclease (New England Biolabs) to a final concentration of 0.1 U/ul or 0.5 U/ul was added to the reactions as indicated and incubated for 15 min at 37 °C. Reactions were terminated by the addition of denaturing stop buffer (90% formamide and 50 mM EDTA). DNA was resolved in denaturing 15% poly-acrylamide gels and imaged at the Cy3 wavelength using FluorChem Q imager (Alpha Innotech). EcoRI endonuclease assay: 5 nM 5′-[32]P radiolabeled pssDNA (RP538/RP539) in buffer (25 mM Tris-HCl, pH 7.5, 1 mM DTT, 0.01% NP-40, 0.1 mg/ml BSA, 10% glycerol, 10 mM MgCl₂, 2 mM ATP, 30 mM NaCl) was incubated with or without indicated Polθ enzyme for 15 min at 37 °C. After initial incubation, 4 units of EcoRI were added as indicated and incubated for 10 min at 37 °C. Reactions were terminated by the addition of non-denaturing stop buffer (100 mM Tris-HCl, pH 7.5, 10 mg/ml proteinase K, 80 mM EDTA, and 0.5% SDS) and incubated at 37 °C for at least 15–20 min. DNA was resolved in non-denaturing 15% polyacrylamide gels and analyzed by phosphorimager (Fujifilm FLA 7000).

**Confocal microscopy**. Glass bottom wells were coated with 30 mg/ml BSA for 30 min at room temp. For foci formation frequency comparisons 40 nM of Cy3 labeled DNA (RP334Cy3) was incubated with 3 nM of the indicated Polθ enzyme. For Rad52 confocal experiments 100 nM Cy3 labeled DNA (RP334Cy3) was incubated with the indicated amounts of Rad52. For time course confocal assays and end-joining confocal assays 15 nM of the indicated Polθ enzyme was incubated with 40 nM DNA (RP344Cy3). Reactions were performed with 25 mM Tris-HCl, pH 8.8, 1 mM DTT, 0.1 mg/ml BSA, 10% glycerol, 10 mM MgCl₂, 2 mM ATP, 30 mM NaCl and 0.01% NP-40. Reactions comparing foci formation frequency were incubated for 10 min at room temp before imaging, excluding time course reactions which were imaged as indicated. Cy3 foci were identified using a Leica DMi8 scanning confocal microscope with a 63x objective at the Cy3 emission wavelength. Random fields were collected for each condition and quantified using

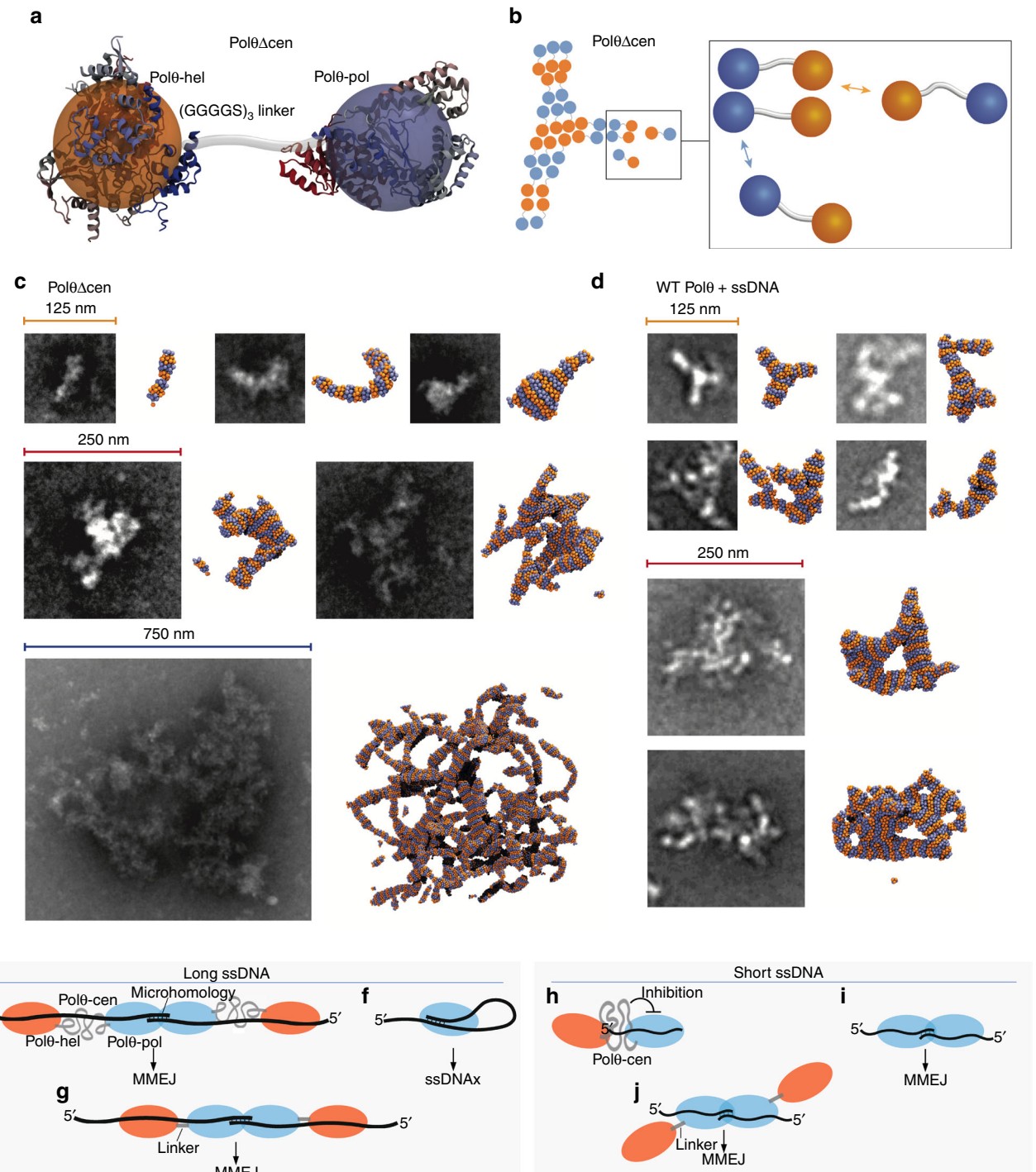

**Fig. 7** Structural and functional models of wild-type Polθ and Polθ mutants. **a** Oversimplified model of PolθΔcen. Polθ-hel and Polθ-pol are represented by two spheres of radius 2.8 nm (the gyration radius of each domain). These spheres are linked by 6 beads of 1 nm in diameter, representing the (GGGGS)$_3$ linker. **b** Typical cluster formation scheme. Polθ-hel is attracted to Polθ-hel (orange with orange), and Polθ-pol to Polθ-pol (blue with blue), while all other bead interactions are repulsive. The PolθΔcen monomers multimerize forming parallel complexes that stack upon each other, see also Supplementary Movies 5 and 6. **c, d** EM images of multimeric PolθΔcen (**c**) and Polθ + ssDNA (**d**) gel-like complexes at different scales alongside complexes of similar shapes obtained from simulations. The simulated molecular structures and EM images are shown using the same magnification. **e** Model of Polθ activity on long (≥ 70 nt) ssDNA. Polθ-hel promotes Polθ-pol MMEJ *in cis* in an ATPase independent manner. **f** Polθ-pol primarily promotes ssDNAx via snap-back replication on relatively long ssDNA due to the absence of helicase attachment. **g** PolθΔcen exhibits identical MMEJ activity to Polθ on long (≥ 70 nt) ssDNA. **h** Polθ-cen suppresses Polθ activity on short ssDNA. **i** Polθ-pol promotes MMEJ of short (≤ 12 nt) ssDNA. **j** PolθΔcen promotes MMEJ on short (≤ 12 nt) ssDNA like Polθ-pol

**ImageJ particle analysis.** Time course reactions were imaged at a single fixed field and quantified over time using ImageJ particle analysis of each individual frame. End-joining reactions performed with confocal imaging were performed as written with the addition of 20 μM dNTPs. After confocal imaging, reactions were terminated by the addition of nondenaturing stop buffer (100 mM Tris–HCl, pH 7.5, 10 mg/ml proteinase K, 80 mM EDTA, and 0.5% SDS) and incubated at 37 °C for 20 min. Reactions were resolved in non-denaturing 12% polyacrylamide gels and imaged at the Cy3 wavelength using FluorChem Q imager (Alpha Innotech).

**Fluorescence anisotropy.** Binding reactions were performed at room temp in 25 mM Tris–HCl, pH 8.8, 1 mM DTT, 0.01% NP-40, 10 mM $MgCl_2$, 10% glycerol, 0.1 mg/ml BSA, 1 mM ATP, 30 mM NaCl for at least 30 min at room temp. Reactions contained 10 nM FAM-conjugated ssDNA (RP316FAM5), and the indicated amounts of the indicated Polθ enzyme. A Clariostar (BMG Labtech) plate reader was used to measure fluorescence anisotropy. All experiments were performed in triplicate, normalized, and plotted with ± s.d.

**Fluorescence Resonance Energy Transfer.** Binding reactions were performed at room temp in 25 mM Tris–HCl, pH 8.8, 1 mM DTT, 0.01% NP-40, 10 mM $MgCl_2$, 10% glycerol, 0.1 mg/ml BSA, 1 mM ATP, 30 mM NaCl for at least 1 hr at room temp. Reactions contained 10 nM each of RP540Cy3 and RP541Cy5 and the indicated Polθ enzyme. A Clariostar (BMG Labtech) plate reader was used to measure FRET (540 nm, excitation; 675 nm, emission). All experiments were performed in triplicate, normalized, and plotted with ± s.d.

**Scanning force microscopy.** Proteins (10 nM) with or without ssDNA (20 nM) were incubated in MMEJ buffer without dNTPs or BSA containing: 25 mM Tris–HCl, pH 8.8, 30 mM NaCl or 100 mM NaCl where indicated, 10 mM $MgCl_2$, 2 mM ATP, 0.01% NP-40 and 1 mM DTT. Incubations were carried out at 37 °C for 30 min and deposited onto freshly cleaved mica. After 15 s the mica surface was washed with milli Q water and dried with a stream of filtered air. Images were obtained on a NanoScope IV SFM (Digital Instruments; Santa Barbara, CA) operating in tapping mode in air with a type J scanner. Silicon Nanotips were from AppNano (Santa Clara, CA). Images were collected at 3 μm × 3 μm and flattened to remove background slope using Nanoscope software. The size of proteins was measured from NanoScope images imported into IMAGE SXM 1.89 (National Institutes of Health IMAGE version modified by Steve Barrett, Surface Science Research Centre, Univ. of Liverpool, Liverpool, U.K.). Statistical analysis was done using QtiPlot (Version 0.9.8.9 svn 2288) and LibreOffice (Version: 5.1.6.2). The volume of proteins monomers, measured from SFM images of proteins deposited in 10 mM Tris–HCl, pH 8.8/100 mM KCl, were 100 $nm^3$ for Polθ-hel and Polθ-pol, 210 $nm^3$ for PolθΔcen and 320 $nm^3$ for Polθ and PolθK121M. These values (± 50%) were used to normalize the volume of observed complexes. Quantification is presented as % of total proteins present in certain oligomeric state.

**Electron microscopy.** Polθ or PolθΔcen (200 nM) with or without ssDNA (2 μM) was incubated in 40 μl of MMEJ reaction buffer without dNTPs, BSA or detergent (25 mM Tris–HCl, pH 8.8, 1 mM DTT, 10 mM $MgCl_2$, 2 mM ATP) at 37 °C for 30 min. Samples (4 μl) were applied to glow discharged lacey carbon grids (Ted Pella, Inc.) and incubated for 2 min at room temp. The grids were then washed with water three times followed by staining with 2% uranyl acetate for 10 s. Excess staining was removed by filter paper. The grids were air dried and imaged in a FEI Tecnai BioTwin Spirit transmission electron microscopy at Penn State University Cryo-Electron Microscopy facility.

**Proteins.** Polθ-pol, Polθ-hel and RPA were purified as described[14]. Polθ, PolθK121M and PolθΔcen were purified as follows. 3xFLAG Polθ, PolθK121M and PolθΔcen GAL1–10 expression plasmids were transformed into yeast strain LSY0269 (a leu2 trp1 ura3-52 prb1-1122 pep4-3 prc1-407 GAL +) or ATCC 208289/BJ5465 (a ura3-52 trp1 leu2-delta1 his3-delta200 pep4::HIS3 prb1-delta1.6 R can1 GAL +), as previously described[35]. Colonies were picked and grown in SC-TRP with 2% raffinose at 30 °C. Starter cultures were expanded and grown to an $OD_{600}$ of 0.6–1.0. Expanded cultures were grown to an $OD_{600}$ of 1.6–2, diluted with YPR to $OD_{600}$ of 0.8–1.0 and induced as follows. Expression was induced by the addition of 2% galactose for 5 hr at 30 °C. Cells were harvested by centrifugation at 6,000 rpm and washed with 50 mM HEPES pH 8.0 and 1 M Sorbitol then washed with lysis buffer (50 mM HEPES, pH 8.0, 1 M NaCl, 10% glycerol, 0.1% Igepal CA630, 1 mM EDTA, 1 mM PMSF, 0.5 mM benzamidine, and Sigmafast inhibitors). Cells were crushed in a freezer mill with liquid nitrogen. Frozen cell powder was stored at −80 °C until purification. Frozen cell powder was thawed and resuspended in lysis buffer. The resuspended cell powder was centrifuged at 92,000 g at 4 °C for 30 min. The clarified supernatant was re-centrifuged at 256,000 × g at 4 °C for 1 h. The supernatant was collected. Anti-FLAG M2 resin was washed with 0.1 M Glycine-HCl, pH 3.5; then with TBS buffer pH 7.4 (50 mM Tris–HCl, 150 mM NaCl) then with lysis buffer by centrifugation at 1000 rpm at 4 °C for 5 min. In total 5 μg/ml 3xFLAG peptide (Sigma) and equilibrated anti-FLAG M2 resin (Sigma) were added to each tube and incubated at 4 °C. The resin was settled by centrifugation at 1,000 rpm at 4 °C for 5 min and flow-through was collected. The resin was washed with lysis buffer and then with wash buffer A

(50 mM HEPES, pH 8.0, 1 M NaCl, 10% glycerol, 0.1% Igepal CA630, 1 mM DTT, 1 mM PMSF, 0.5 mM benzamidine,10 mM $MgCl_2$). The resin was incubated in wash buffer A on ice for 15 min, then the resin was settled by centrifugation at 1,000 rpm at 4 °C for 5 min. The resin was then washed with wash buffer B (50 mM HEPES, pH 8.0, 1 M NaCl, 10% glycerol, 0.1% Igepal CA630, 1 mM DTT, 1 mM PMSF, 0.5 mM benzamidine). The resin was resuspended in elution buffer (50 mM HEPES, pH 8.0, 1 M NaCl, 10% glycerol, 0.1% Igepal CA630, 1 mM DTT) with 500 μg/ml 3xFLAG peptide (Sigma) at 4 °C with rotation. A disposable 10 ml polypropylene column (Thermo Fisher) was washed with TBS buffer, pH 7.4 (50 mM Tris–HCl, 150 mM NaCl) and then with elution buffer. The resin was loaded to the column and elution fractions were collected. The elution was dialyzed against dialysis buffer (50 mM HEPES, pH 8.5, 250 mM NaCl, 10% glycerol, 0.1% Igepal CA630, 1 mM DTT, 7.5 mM ATP) overnight at 4 °C. Polθ, PolθK121M and PolθΔcen concentrations were determined by SDS gel analysis and by specific activity using Polθ-pol as a standard. Polθ, PolθK121M and PolθΔcen were stored in aliquots at −80 °C.

**DNA.** Templates are as follows: Figs. 1f, 4d, i RP469D/RP486; Figs. 1i, k, 2b, 4e, j, l SJB108; Figs. 1j, l, n, o, q, 2a, 3d, h, 4f, k SJB116; Fig. 1p left, 4 M, 4 N RP344; Fig. 2f, RP514; Fig. 2j, RP515; Fig. 2k, RP516; Fig. 2h, RP132; Fig. 2i, RP132C. Figure 3e, SJB116/RP343; Fig. 3f, SJB116/RP344/SJB155/SJB154; Fig. 3g, SJB116/RP344/SJB155/SJB154/SJB153; Fig. 3i SJB116/RP343; Fig. 3j SJB116/ SJB158; Fig. 3k SJB116/SJB159; Fig. 5a RP540Cy3/RP541Cy5; Fig. 5d RP538/539.

Primer templates were 5′- phosphorylated on the shorter strand with T4 polynucleotide kinase (New England Biolabs) and ATP. Primer templates were annealed by mixture of a ratio of 1:2 of short to long strands then boiling and slow cooling to room temp. DNA was $^{32}$P-5′-radiolabeled with T4 polynucleotide kinase (New England Biolabs) and (γ-$^{32}$P) ATP (Perkin Elmer). RP469D: CTGTCCTGC ATGATG;RP486: CACTGTGAGCTTAGTCACATTTCATCATGCAGGACAG;R P344: CACTGTGAGCTTAGGGTTAGCCCGGG;RP348: CACTGTGAGCTTAG GGTTAGAGCCGG;SJB108: GTTCTTCGGTCTCGAGGTGACTACAAGGATGA CGACGACAAGGGCACTGTGAGCTTAGGGTTAGCCCGGG;SJB116: CACTG TGAGCTTAGGGTTAGGCGGCTTGCAGAGCACAGAGGCCGCAGAATGTGC TCTAGATTCCGATGCTGACTTGCTGGGTATTATATGTGTGCCCGGG;RP5 14: GTTCTTCGGTCTCGAGGTGACTACAAGGATGACGACGACAAGGG CACTGTGAGCTTAGGGTTAGAAATTT;RP515: GTTCTTCGGTCTCGAGG TGACTACAAGGATGACGACGACAAGGGCACTGTGAGCTTAGGGTTAG CCGG;RP132: GACGTTGACTTAAAGTCTAACCTATAGGATACTTACAG CCATCGAGAGGGAGCACGGGCGATTCTCGAGCGTAC;RP132C: GCTCGAG AATGCGCCGTGTCCCTCTCGATGGCTGTAAGTATCCTATAGGTTAGACTT TAAGTCAACGTCGTAC;RP370: CACTGTGAGCTTAGGGTTAGGAATTC;RP 316FAM5: /56-FAM/TTTTTTTTTTTTTTTTTTTTTTTTTTTTTTT;SJB123: GGTTA GCCCGGG;RP344Cy3: /5Cy3/TTTTTTTTTTTTTTTTTTTTTTTTTTTTTTT TTTTTTTTTTTTTTT;RP343: CTAAGCTCACAGTG;SJB153: TATAATACCCA GCAAGTCAGCATC; SJB154: GGAATCTAGAGCACATTCTGCGGCC;SJB155: TCTGTGCTCTGCAAGCCGCCTAACC;SJB158: TATAATACCCAGCAAGTCA GCAT/3ddC/;SJB159: GGAATCTAGAGCACATTCTGCGGC/3ddC/;RP540Cy3: GCATATTCACTGTGAGCTTAGTGTTAGGACTCGG/3Cy3Sp/;RP541Cy5: GC ATATTCACTGTGAGCTTAGTGTTACCGAGTCC/3Cy5Sp/;RP538: CGACAAG AGTCATGAATTCTTAGGGTTAGCCCGGG;RP539: CTAAGAATTCATGACT CTTGTCG;SJB089: CAATTCAGCAACTAATGTCATACCAGCTGAAGTTGG TGCAGAG;SJB090: CTCTGCACCAACTTCAGCTGGTATGACATTAGTTGCT GAATTG;SJB100: TTCTGCTAGCGGTGGTGGAGGAAGTGGAGGAGGC GGATCTGGTGGTGGCGGTAGCGGTTTTAAAGATAACTCTCCAATCTCAG ATACTTC;SJB101: TCTTCTGCTAGCCATTTCAACCAAATCTTGTTGCAAG.

**Plasmids.** 3xFLAG-Polθ plasmid was derived from pRS424 (ATCC). Human Polθ nucleotide sequence was optimized for yeast and synthesized as 2 gene fragments by GenScript. Yeast optimized gene fragments were cloned into pRS424 sequentially. 3xFLAG-PolθK121M plasmid was derived from 3xFLAG-Polθ plasmid by site-directed mutagenesis using primers (SJB089 and SJB090). 3xFLAG-PolθΔcen was derived from 3xFLAG-Polθ plasmid by molecular cloning with PCR primers (SJB100 and SJB101).

**Western blot.** Polθ, Polθ-pol, and PolθΔcen were resolved by SDS/PAGE and transferred to a nitrocellulose membrane. The membrane was blocked with 5% milk in TBS with 0.1% Tween-20 and incubated with primary antibody against C-terminal portion of Polθ (rabbit anti-Pol theta; ThermoFisher #PA5-69577) in TBS/5% milk/0.1% Tween-20 for 1.5 hr at room temp, followed by incubation with HRP-conjugated secondary antibody (Goat Anti-Rabbit IgG; Upstate #12-348) for 1 h at room temp. Bands were detected via chemiluminescence.

**Modeling of PolθΔcen using crystal structures.** The PolθΔcen:ssDNA model was constructed by using the previously determined crystal structures of human Polθ-hel (PDB, 5AGA)[13] and Polθ-pol (PDB, 4 × 0 P)[16]. The Polθ-hel:ssDNA model was constructed by superposing Polθ-hel structure with the crystal structure of archaeal DNA helicase (Hel308):DNA complex[28], and the strand in which Hel308 translocates on was included in the model. The template DNA in the Polθ-pol:DNA complex was removed and the 5′ end of the primer DNA was connected

to the 3′-end of DNA in the Polθ-hel model. Polθ-hel was modeled facing upstream due to its 3′−5′ translocation activity.

**Computational modelling for PolθΔcen.** PolθΔcen is modelled by two large spheres, representing the polymerase (P) and helicase (H) domains, connected by a short flexible polymer chain made by 6 beads, representing the $(GGGGS)_3$ linker (L), see Supplementary Fig. 7M. Each bead L represents about two and half amino acids, and have a mass 300 times smaller than P and H.

Each bead is governed by a Langevin equation of motion at temp $T = 300$ K and with a potential energy $V$, composed by the following interaction terms.

The interactions between polymerase-polymerase and helicase-helicase are attractive, and are modelled with a Lennard-Jones potential $V_{P-P}(r) = V_{H-H}(r) = \varepsilon$ $((\sigma/r)^{12}-(\sigma/r)^6)$, where $\sigma = 5.6$ nm is about twice the gyration radius of both P and H and $\varepsilon = 10K_BT$ is the interaction strength. The potentials $V_{P-P}$ and $V_{H-H}$ are truncated at $r = 8$ nm, which correspond to a cutoff distance of about 1.5 nm from their potential minimum, situated at $2^{1/6}\sigma$. The distance 1.5 nm corresponds roughly to the Debye screening length for this system.

All other bead-bead interactions are instead only repulsive and modelled with a Lennard–Jones potential truncated at the potential minimum $r = 2^{1/6}\sigma$.

$\sigma$ and $\varepsilon$ vary depending on the particle types. For polymerase-helicase interactions $(V_{P-H}(r))$, $\sigma = 5.6$ nm and $\varepsilon = 10K_BT$. For linker-linker interactions $(V_{L-L}(r))$, $\sigma = 1$ nm, corresponding to about 2.5 amino acids, and $\varepsilon = 1K_BT$. For polymerase-linker and helicase-linker $(V_{P-L}(r) = V_{H-L}(r))$ $\sigma = 3.3$ nm and $\varepsilon = 10K_BT$.

The connectivity between the beads of each PolθΔcen model (as shown in Supplementary Fig. 7M) is implemented with harmonic potentials of the form $V_H = K(r-r_0)$, where $K = 750$ $K_BT/nm^2$ and $r_0 = 1$ nm for linker-linker bonds and $r_0 = 3.3$ nm for linker-helicase and linker-polymerase bonds.

To generate the assemblies of varying size, we initialized the system with proteins randomly distributed at densities of about $1–3 \times 10^{-4}$ proteins/nm$^3$ and cubic boxes of size between 100–600 nm. The chosen densities, about $10^4$ times larger than the experimental one ($\sim 10^{-8}$ proteins/nm$^3$), and the size range of the simulation boxes allow to reproduce the local concentration, size and shape of multimers observed in the EM data.

The simulations are carried out with the software LAMMPS with reduced units $\sigma = 1$ nm, $\varepsilon = 1K_BT$ and m = 900 Da. In simulation units, the integration timestep is dt = 0.01 and the friction coefficient is $\gamma = 0.5$. The simulation box is bounded by repulsive walls with a repulsive truncated Lennard–Jones potential, where the truncation is set at $r = 2^{1/6}\sigma$. A typical simulation is long $5–6 \times 10^6$ time units.

The clusters generated by the self-assembly process have a peculiar elongated shape, which is also characteristic of the EM images. This structural feature is absent if the helicase-helicase or polymerase-polymerase attractive interactions are substituted with a truncated repulsive Lennard-Jones potential, see Supplementary Fig. 7N. Moreover, clusters assembled with an additional attractive force between Helicase and Polymerase domains can still form elongated shapes, but the phenomenon is less pronounced, see Supplementary Fig. 7N. For these reasons, we believe that the selective attractions between P-P and H-H are a fundamental driving force for the assembly of the protein in an elongated form.

**Reporting Summary**. Further information on research design is available in the Nature Research Reporting Summary linked to this article.

## Data availability
The generated data that support these findings are available in the Source Data file provided and in the Supplementary Information. Any additional data is available on reasonable request from the corresponding author.

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

## Acknowledgements

This study was supported by the National Institutes of Health, grants to R.T.P. (1R01GM115472-01, 1R01GM130889-01), T.S. (1R01CA186238), and K.S.M. (R01GM087350). We are grateful to Drs. L. Prakash and S. Prakash for providing Polθ-pol purified from yeast. This research includes calculations carried out on Temple University's HPC resources and thus was supported in part by the National Science Foundation through major research instrumentation grant number 1625061 and by the US Army Research Laboratory under contract number W911NF-16-2-0189.

## Author contributions

S.J.B., A.Y.O., T.R., T.K., T.H., D.R. and Y.S. performed the experiments and interpreted the data. E.K. developed the methods for Polθ purification. S.J.B., A.Y.O., T.K. and T.H. purified the proteins. K.S.-R. and T.S. were responsible for the Western blots. A.S. and V.C. designed and implemented the computational modeling studies and interpreted the data. C.W. and K.S.M. were responsible for overseeing the molecular imaging studies. K.S.M. generated the structural model of PolθΔcen. T.R. and G.C. sequenced and analyzed MMEJ products. S.J.B., A.Y.O., T.K., T.R., C.W., K.S.M., A.S. and V.C. provided editorial input. T.R., L.S., N.B. and J.S.M. contributed to data collection, processing and quantitation. R.T.P. designed the study, directed the research, interpreted the data, and wrote the manuscript.

## Additional information

**Competing interests:** R.T.P. has filed patents related to this study. Remaining authors declare no competing interests.

