## [Peer Review File · Nature Communications]

Reviewers' comments:

Reviewer #1 (Remarks to the Author):

DNA repair pathway choices are important for disease development and critical determinants of cancer therapy response. In the presented manuscript, Black et.al. describe unexpected and so far elusive functions of both the disordered central domains and the helicase domains of POLQ, a TLS polymerase that is acting during micro-homology mediated endjoining (MMEJ). Specifically, the helicase domain independently of ATP is required for MMEJ with long ssDNA, and facilitates DNA synapsis and so MMEJ. The disordered central region inhibits MMEJ of small ssDNA substrates and inhibits protein aggregation in the absence of DNA. Both domains increase ssDNA binding in vitro.

MMEJ in cells preferentially occurs on long ssDNA overhangs. Yet, the polymerase domain of POLQ is incapable of performing such reactions in vitro. The authors present a lot of data to convincingly establish that the hel domain is required for MMEJ reactions involving long ssDNA, specifically by suppressing intramolecular snap-back reactions by the polymerase and for DNA synapsis. The central domain on the other hand appears not essential for long ssDNA MMEJ (Fig. 5K) but rather the data nicely reveals an unexpected autoinhibitory function for Pol θ -cen on short ssDNA, thus limiting POLQ activity and, contributing to substrate specificity and so ultimately pathway choice.

Overall, some elegant assays are given and the majority of the data is convincing. However, the layout of the manuscript and figures is in part confusing and does not serve the take-home points, some of the interpretation are one-sided and some of the conclusions too speculative as currently written. The observation of gel-like oligomerization is very interesting, however at this point a bit premature and not necessary for the main conclusions and their impact of this manuscript.

Here are some major points to consider for improving the manuscript:

1) The manuscript could be extensively shortened by a succinct introduction with a clear delineation of the salient missing understanding of POLQ, and by eliminating non-essential and duplicate information about reactions conditions also found in the method section. Switching between Figures should be avoided (describing Fig 1, but referring to Fig.4, or in the text describing Fig. 1A, B, C, then jumping to Fig J, which appears to be the same data replotted, then going back to Fig. 2G, then going back to Fig 2D), and all Figure panels should be mentioned in the text (e.g. multiple panels in Fig. 2 seem not to be mentioned). Figure 7, if included, should be part of the results, not discussion. Overall, it is somewhat unnecessarily difficult to follow the manuscript and the conclusions, which masks the importance of the conclusions. More clarity easily could make the implications of greater interest to a general audience. Another example for one-sided representation is that reported cellular and biochemical properties of POLQ are often being mixed for arguments, but no consideration for importance outside MMEJ is given. e.g the authors suggest that "Recent cellular studies, however, indicate that Pol θ -cen and Pol θ -hel are dispensable for Pol θ TLS activity on primer templates²⁰. This suggests the possibility that Pol θ -cen and Pol θ -hel may exclusively function in MMEJ by cooperating with Pol θ -pol." POLQ could have more functions in cells than TLS and MMEJ- e.g. POLQ is also found in the mitochondria with yet unresolved functions.

2) The structural modeling of POLQ pol and POLQ hel is speculative, as it is a model (POLQ hel) of a model over a model of a different model (Hel308), artificially linked to a third model (DNA of POLQpol) and therefore should not be used as a supporting data for their interpretation, but rather as a model for their actual data. The authors conclude that "The model supports the ability of Pol θ -hel to strongly contribute to Pol θ ssDNA binding". This cooperativity should be supported by the binding kinetics seen in Fig 4A: the binding curves for pol and hel domains are only partially given. Where do the curves saturate? This is not only important for accurate KD calculations, but also for further interpretation: Sigmoidal binding curves suggest cooperativity, as opposed to single

hyperbolas. How do the authors explain a cooperative binding of POLQ on DNA? On another note for graphical representation, why / how can the POLQ curve start at -0.5 nM? Is the X-axes mislabeled?

3) The authors suggest that the ATPase activity of the helicase domain is not required, using an ATP binding mutant K121M as a proof of principle. However, the authors never show if this mutant is defective in ATP hydrolysis or ATP binding or both. If this information is unknown, the text should be modulated accordingly. There are many examples that show ATP binding but not hydrolysis to be required for multimeric protein assembly (e.g. the proteasome <https://www.ncbi.nlm.nih.gov/pubmed/17018291>.) The high amounts of ATP needed for stability of the protein during purification could hint on ATP as a structural co-factor, rather than an enzymatic requirement. Thus, if the mutant retains ATP-binding, it may be sufficient for its biological function. While the authors nicely demonstrate that POLQ-hel promotes inter-molecular pairing by suppressing intra-molecular reactions, it would be insightful if the authors could discuss the requirement for a helicase domain in this process - even if the translocation activity requiring ATP hydrolysis is not required.

4) The interpretations with regards to the gel-like aggregates and the function of the POL-cen domain in this context leave many questions unanswered with alternative interpretations: The authors conclude that the major mechanistic function of Polθ-cen is to tether Polθ-hel to Polθ-pol (Fig.4B). However disordered regions play important roles for phase separation reactions, while "gelling" is associated so far with pathological protein aggregates, such as accumulated during Alzheimers disease ([https://www.cell.com/trends/cell-biology/references/S0962-8924\(18\)30028-X](https://www.cell.com/trends/cell-biology/references/S0962-8924(18)30028-X), <https://www.biorxiv.org/content/10.1101/621714v1>) What is the evidence of gel-like aggregates versus fluid aggregates? Can the authors comment on the dissolution of these gel-like aggregates? Indeed, the authors show that in the absence of DNA, the POL-delta-cen dramatically increase aggregates compared to wild-type, which may be pathogenic, in a context where no pairing is wanted, e.g. in the absence of ssDNA overhangs. Thus, the conclusion that the cen domain solely acts as a linker may be premature. E.g. aside the number of proteins in the oligomerized aggregates, is there any quality difference between the aggregates with and without DNA, and especially in the aggregates with and without the cen domain? Are the aggregates more or less fluid?

Alternatively, it seems that the authors show much data and insights on POLQ domain functions for MMEJ and pathway choice. The aggregation/ gelling is very interesting but perhaps could benefit from a more thorough, independent follow up study, and could be removed from this manuscript without taking away from the novelty and insight that Figs 1-5 provide.

Minor points:

The authors use long ssDNA substrates in the absence of RPA or other ssDNA stabilizing proteins to demonstrate that long ssDNA are needed for POLQ-MMEJ

Can the authors exclude that the ssDNA forms secondary structures and the ssDNA length requirement could reflect a need for dsDNA? This also could explain why full length POLQ is active on a p/tp construct even with short ssDNA overhang (Fig. 1F)

If the ssDNA_x products in the gel are created from snap-back reactions, why are they not cleaved by XmaI, since it is also initiated by the same CCCGGG sequence. This could be clarified in the text.

A lot of kinetic measurements are presented and it would be helpful to add the actual rate values, e.g. in ATPase figures of Fig 1G, or MMEJ rate of full length POLQ vs POLQ-delta-cen (fig 1N, 4K, and 4F) etc. This would help in comparing and in the actual description of the polymerase

properties.

What is the difference between Fig. 2J and A-C? In fact, it appears that the data was replotted judging from the error bars?

Define "POL cen" on p4

Define pssDNA

Reviewer #2 (Remarks to the Author):

The paper entitled "Molecular Basis of Microhomology-Mediated End-Joining by Purified Full-Length Polθ" described their work on the comprehensive in vitro biochemical analysis of full-length and different truncation proteins of recombinant human Polθ (I use polθ because it's easier to type) to elucidate the specific roles of Polθ-hel, Polθ-pol, and Polθ-cen in MMEJ using various lengths of ssDNA substrates, and to determine interplays of these subdomains for the function of Polθ. Besides its critical role for DNA repair, it has important therapeutic value for cancer through the synthetic lethality mechanism. The data are well presented. The findings are novel, and conclusions are well supported by the abundant data on this important large DNA transaction enzyme whose soluble protein was difficult to obtain for a clean and comprehensive in vitro functional characterization. I support its publication in Nature Communication, with the following minor suggestion for revision.

1). Full length polθ and polθ-dcen formed gel like oligomeric aggregates, which is a very intriguing proposal, and makes me wonder about the following to questions for the author to consider.

a). Could such a gel-like aggregates function as a storage site of polθ inside nucleus? Does polθ show some subcellular localization, in some sort of granular-structure in cells as a storage site for fast access when needed ?

b). Because of its ability to oligomerize on dsDNA, could polθ play a role in chromosome organization also associated with dsDNA break damaging and repair ?

2). What's the evidence of the central domain to be unstructured besides the disorder prediction? If really unstructured for such a long stretch (over 800 aa), it should not be stable (will be degraded inside the cells) or tangled with itself or other proteins. More likely it have extended structure to behave as an elongated and flexible domain. Maybe "flexible connector" or similar sort is a better term than unstructured domain for polθ-cen?

3). Define hydrogen bonding potential? (similar to electrostatic potential or number of H bonds?)

4). Fig. 1A, indicate the aa number at the beginning and end of polθ-dcen region.

5). Fig. 6. Indicate ssDNA length used in legends for the different panels so that readers do not have to hunt that down in the main text.

6). Indicate the length of dsDNA used in the figure legend in S-Fig. 7.

7). S-Fig. 7, in panels G-L, it appears polθ also bind dsDNA and form large aggregates. What's the binding affinity of polθ to dsDNA (for comparison to ssDNA binding). This affinity comparison may

help understand the function of polQ in cells.

8). The model predicts the helicase binding to the 5' side of the ssDNA and pol to the 3' side. It follows that the 3'↔5' translocation of helicase will allow the helicase to move further to the 5' end, away from pol at the 3'-end, fully extend the cen linker. This would help increase the binding to ssDNA if the initial binding through helicase domain happen to be at the 3'-end and pol domain not binding to ssDNA, as translocation from 3'↔5' would allow binding of pol to the vacant 3' ssDNA.

Reviewer #3 (Remarks to the Author):

Pol theta is known to be important to genomic integrity, and thus to human health and disease. It is a very large protein (290 kDa) containing a N-terminal helicase and a C-terminal polymerase, connected by a disordered central region. It is unknown whether the helicase domain or central region influence or communicate with the Pol domain, because previous work by many labs have, to date, have only studied the Pol domain for lack of obtaining the full length protein. This report by Samuel Black et. al. studies the full length Pol theta, which they succeeded to obtain by using yeast as the expression vector. While the Pol domain alone can perform TLS (translesion synthesis) and end-joining of short 3' overhangs, we learn from this report that the full length helicase-Pol of Pol theta has significantly different properties from the isolated Pol domain. Notably, the full length protein is uniquely capable of microhomology mediated end joining (MMEJ) of long 3' overhangs, thereby reconstructing the physiological MMEJ reaction of Pol theta documented to occur in the cell. Furthermore, the authors show that the helicase activity per se is not required (i.e. ATP active site mutant is active), yet the helicase domain, a SF2 helicase (related to RecQ helicases), is necessary for MMEJ. The authors demonstrate that binding of the helicase domain to DNA likely prevents intrastrand snapback pairing within long 3' ssDNA overhangs which, left unchecked, inhibits MMEJ. The authors also show that the length of the disordered central region can be substituted for a short linker without disrupting MMEJ activity, and thus the cen region is not directly involved, but is required to link the helicase and Pol in cis for the observed MMEJ activity, consistent with their observations that addition of the Pol and Helicase domain in trans do not perform the reaction.

The reconstitution of the MMEJ reaction with long 3' overhangs is an important step in understanding the essentially synthetic lethality of Pol theta in combination with BRCA1/2 deletions. Specifically, normal DSB repair by high fidelity recombination is accommodated by the error-prone MMEJ activity performed by Pol theta, or Ku mediated non homologous end joining. However, the Ku pathway can only repair DNA breaks with short 3' overhangs, thus leaving the mechanism of repair of longer 3' overhangs a mystery.

The report is well written, the data are fully documented, and the conclusions follow from the data. The report will appeal to a large audience, and I fully expect will garner many citations. In overview, I have no major comments and believe the report is of wide interest in the major fields of repair, recombination and replication. Minor comments that the authors may consider are listed below.

Minor Comments:

1) Given the expected orientation of Pol and helicase, modeled in Fig. 5, would the authors have examined whether the helicase activity is needed in the presence of ssDNA binding factors like Rad51 or RPA? This might be part of the authors future plans, but if this has been done already, it would help the reader understand if the helicase plays a role in this reaction. For example, the way it reads now – it seems that the helicase might be involved in a distinct molecular pathway.

2) If the helicase domain were to mainly be a dsDNA translocase instead of a regular bona fide helicase, it might translocate along dsDNA for the purpose of rapidly targeting the Pol domain to

an end, while at the same time binding ssDNA of long 3' overhangs to suppress intramolecular snapback. Have the authors examined the FL Pol theta for helicase activity in simple helicase assays?

3) The authors might consider in the future to try experiments comparing the ATPase mutant vs wt FL Pol theta on substrates containing different lengths of dsDNA? If the helicase domain is mainly for dsDNA translocase action (e.g. like SMARCAL1 etc), the ATP proficient enzyme may enable more rapid binding/location of Pol theta to the 3' ssDNA end, as assayed using the restriction enzyme protection assay developed in this report.

4) The authors may want to consider (in future studies, not this one) cross-linking mass spectrometry of FL Pol theta +/- ssDNA, to map how extensive of an interface, if any, exists between the helicase and Pol domains. This would be informative to see if the helicase-Pol communication is direct, or indirect, and whether there are significant changes upon binding to a long 3' ssDNA overhang.

5) The Pol theta lacking the cen region appears as capable of MMEJ as the full length protein. This suggests, as the authors note, that the cen domain is not essential. However, it would be informative to test the reaction in a NaCl titration, comparing wt Pol theta with Pol delta cen, to determine if they still contain similar activity when challenged by ionic strength (or by competitor ssDNA).

6) It became a bit confusing to read that the cen region is not required for MMEJ, yet to also read that the central region is demonstrated to be required to restrict the MMEJ activity to long 3' overhangs, and down regulates MMEJ of short 3' overhangs. Maybe it was just me. But if other reviewers make the same comment, perhaps a bit of clarification on what the authors are thinking regarding the cen region.

7) The following comment does not need an experiment to be performed for the current publication, but I only mention it out of my own curiosity.

Regarding the multimeric status of Pol theta in the reaction, and the question of whether large multimers of the enzyme are needed, would this assay be amenable to the separation of the reaction into two steps such that only one end of each of two substrates with micro homology is preincubated with Pol theta, and then to remove excess Pol theta (eg gel filtration) before mixing and examining the MMEJ reaction? Likewise, to see if preinc of both substrates, separately (and gel filtered separately) is sufficient for MMEJ. While this may or may not change the multimerization conclusion, it might determine if the multimerization leading to active MMEJ requires both ends to be simultaneously present in the same reaction to form the proper network for joining.

Signed: Reviewer is Mike O'Donnell (please feel free to disclose this to the authors)

Reviewer 1:

...MMEJ in cells preferentially occurs on long ssDNA overhangs. Yet, the polymerase domain of POLQ is incapable of performing such reactions in vitro. The authors present a lot of data to convincingly establish that the hel domain is required for MMEJ reactions involving long ssDNA, specifically by suppressing intramolecular snap-back reactions by the polymerase and for DNA synapsis. The central domain on the other hand appears not essential for long ssDNA MMEJ (Fig. 5K) but rather the data nicely reveals an unexpected autoinhibitory function for Pol θ -cen on short ssDNA, thus limiting POLQ activity and, contributing to substrate specificity and so ultimately pathway choice.

Overall, some elegant assays are given and the majority of the data is convincing. However, the layout of the manuscript and figures is in part confusing and does not serve the take-home points, some of the interpretation are one-sided and some of the conclusions too speculative as currently written. The observation of gel-like oligomerization is very interesting, however at this point a bit premature and not necessary for the main conclusions and their impact of this manuscript.

Authors response:

We are grateful for the fair and thorough review and appreciate the referee's expertise in enzymatic mechanisms. We agree that the layout of the manuscript can be more cohesive, and that some of the conclusions may be somewhat speculative as currently written. Overall, we strongly agree with the majority of the referee's comments and suggestions and provide point-by-point responses below as to how we addressed the referee's suggestions. In regard to the observations of large oligomerization of Pol θ , and especially Pol θ Δ cen mutant protein which is highly active in MMEJ and thus soluble, we believe that these particular data provide important insight into how this protein behaves at the structural level and partly explains its ability to join together DNA ends. For example, even prior to investigating the multimerization potential of Pol θ in the manuscript, our biochemical data provide evidence that at least two molecules of Pol θ are involved in MMEJ (see Fig. 5D-G). Thus, the logical flow of the manuscript after this point is to investigate whether Pol θ acts as a monomer or multimeric complex to perform MMEJ.

We were quite surprised to find that Pol θ behaves mostly as a monomer and dimer in the absence of DNA, yet forms multimers on DNA. Our low resolution structural studies therefore significantly advance our understanding of how the full-length Pol θ protein behaves at the molecular level. Oligomerization of end-joining proteins is precedented in the literature and may therefore be a common mechanism of DNA synapse formation. For example, RAD52 also forms large multimeric complexes and facilitates a related form of DNA end-joining (single-strand annealing) that requires a certain amount of homology between 3' ssDNA overhangs. We believe that our consistent data using EM, AFM, and confocal microscopy methods that show the tendency of Pol θ to form large complexes on DNA should remain in this first structure function study of full-length Pol θ since no structural data is available on this full-length protein in the literature. Based on the other two reviewers' responses, there are no other objections to including these intriguing Pol θ oligomerization data in the manuscript. For example, reviewer 2 mentions that, "Full length polQ and polQ-dcen formed gel like oligomeric aggregates, which is a very intriguing proposal,..."

Another important finding which is presented as structural data is that replacement of the central domain with the short flexible linker results in the Pol θ Δ cen protein forming large complexes even in the absence of DNA. These structural data therefore reveal a role for the central domain in suppressing Pol θ oligomerization in the absence of DNA which we feel is another important finding in this structure function study. Without the confocal microscopy, AFM and EM data we would have to completely remove this intriguing finding which highlights the central domain as a regulatory domain. We note that we have preliminary cellular data that shows the ability of Pol θ to form large complexes. However, further confirmation and characterization of these data will require a complete follow-up paper. Below, we present point-by-point responses to each of the reviewer's comments. Overall, we believe the revised manuscript takes into account all of the referee's comments, and is more clear and straight forward to follow.

Reviewer 1:

Here are some major points to consider for improving the manuscript:

1) The manuscript could be extensively shortened by a succinct introduction with a clear delineation of the salient missing understanding of POLQ, and by eliminating non-essential and duplicate information about reactions conditions also found in the method section.

Authors response:

We agree with the referee. We have significantly shortened the introduction and focus on the point that it remains unclear how full-length Pol θ (Pol θ) functions on long 3' ssDNA overhangs to promote MMEJ. We have also slightly shortened some of the main text in accordance with *Nat Comm* word limit. We also removed detailed methodology from the main text which is easily found in the methods section.

Reviewer 1:

Switching between Figures should be avoided (describing Fig 1, but referring to Fig.4, or in the text describing Fig. 1A, B, C, then jumping to Fig J, which appears to be the same data replotted, then going back to Fig. 2G, then going back to Fig 2D), and all Figure panels should be mentioned in the text (e.g. multiple panels in Fig. 2 seem not to be mentioned). Figure 7, if included, should be part of the results, not discussion.

Authors response:

We agree with the referee. Thus, we have now avoided switching between figures and all figure panels are mentioned in the text in a more cohesive manner. We have also moved the Figure 7 data from the discussion to the main text as suggested by the referee.

Reviewer 1:

Overall, it is somewhat unnecessarily difficult to follow the manuscript and the conclusions, which masks the importance of the conclusions. More clarity easily could make the implications of greater interest to a general audience. Another example for one-sided representation is that reported cellular and biochemical properties of POLQ are often being mixed for arguments, but no consideration for importance outside MMEJ is given. e.g the authors suggest that "Recent cellular studies, however, indicate that Pol θ -cen and Pol θ -hel are dispensable for Pol θ TLS activity on primer templates²⁰. This suggests the possibility that Pol θ -cen and Pol θ -hel may exclusively function in MMEJ by cooperating with Pol θ -pol." POLQ could have more functions in cells than TLS and MMEJ- e.g. POLQ is also found in the mitochondria with yet unresolved functions.

Authors response:

We again thank the referee for their careful review of the manuscript. We now believe that the manuscript is much easier to follow with the main implications highlighted for a more general audience to understand. We note that the complexities of our findings on this unique dual enzymatic protein were indeed difficult at times to summarize. Based on the referee's comments, we now include the following statement about the possibility that the helicase may function during other unresolved functions of Pol θ in interstrand crosslink repair and mitochondrial DNA replication and repair.

"Pol θ ATPase activity may be involved in an auxiliary DNA repair function, such as DNA unwinding or dissociation of protein-ssDNA complexes^{9,13,15}. The ATPase activity may also contribute to unresolved functions of Pol θ such as in interstrand crosslink repair and mitochondrial DNA replication and repair^{25,26}."

We note that the sentence presented above in quotes by the referee above has been omitted due to shortening the introduction as suggested.

Reviewer 1:

2) The structural modeling of POLQ pol and POLQ hel is speculative, as it is a model (POLQ hel) of a model over a model of a different model (Hel308), artificially linked to a third model (DNA of POLQpol) and therefore should not be used as a supporting data for their interpretation, but rather as a model for their actual data.

Authors response:

We agree with the referee and again thank them for their insight and careful review of the manuscript. We have now removed or re-worded sentences that suggested the data support the model, and instead simply use the model to show support for our empirical data. For example, we now have re-worded some of the text referring to the model as follows:

“Although this model is speculative, it suggests bivalent ssDNA binding by the fusion protein which would be expected to be in the low nanomolar range....”

Taken together, the nuclease protection and FRET assays in Figs. 5B-5D is supported by the Pol θ Δ cen:ssDNA model depicted in Fig. 5A.

Reviewer 1:

“The model supports the ability of Pol θ -hel to strongly contribute to Pol θ ssDNA binding”. This cooperativity should be supported by the binding kinetics seen in Fig 4A: the binding curves for pol and hel domains are only partially given. Where do the curves saturate? This is not only important for accurate KD calculations, but also for further interpretation: Sigmoidal binding curves suggest cooperativity, as opposed to single hyperbolas. How do the authors explain a cooperative binding of POLQ on DNA? On another note for graphical representation, why / how can the POLQ curve start at - 0.5 nM? Is the X-achses mislabeled?

Authors response:

We have now repeated the fluorescence anisotropy assays for both Pol θ -hel and Pol θ -pol using saturating concentrations of the proteins as requested by the referee. The curves now show saturation and the KD calculations are more accurate based on these additional data points. The new curves are now included as Figure 4A. We note that the original fluorescence anisotropy curves were presented as logarithmic scale which made the data plot appear as sigmoidal. The new data are now plotted as a linear X-axis scale (nM protein) which shows no evidence of sigmoidal curves, and thus does not support cooperative binding by the proteins to ssDNA. The X-axes have now been corrected on the new plots shown in Figure 4A.

Reviewer 1:

3) The authors suggest that the ATPase activity of the helicase domain is not required, using an ATP binding mutant K121M as a proof of principle. However, the authors never show if this mutant is defective in ATP hydrolysis or ATP binding or both. IF this information is unknown, the text should be modulated accordingly.

Authors response:

We again thank the referee for their insight and close review of the manuscript. Based on the reviewer's comment, we have now tested ATPase activity for the Pol θ K121M mutant protein and show that this protein fails to promote ATP hydrolysis. These new control data are now added to Supplementary Fig. 3H. We note that this particular residue (K121) within the Walker A motif of the helicase is absolutely conserved in SF2 helicase members and is well documented in its role in binding ATP (Structure. 2015 Dec 1;23(12):2319-2330. doi: 10.1016/j.str.2015.10.014; J Biol Chem. 2018 Apr 6;293(14):5259-5269. doi: 10.1074/jbc.RA117.000565; *Curr Opin Struct Biol.* 2010 Jun;20(3):313-24. doi: 10.1016/j.sbi.2010.03.011; *Front Biosci* (Landmark Ed). 2012 Jun 1;17:2070-88.). As direct evidence for this, a recent structural paper on Pol θ -hel reveals K121 binding to the phosphate backbone of ATP (Structure. 2015 Dec 1;23(12):2319-2330. doi: 10.1016/j.str.2015.10.014), and our recent biochemical

paper on Polθ-hel demonstrates that the Polθ-hel K121M mutant is unable to perform DNA unwinding in an ATP-hydrolysis dependent manner compared to WT Polθ-hel which is fully active in DNA unwinding and ATP hydrolysis (J Biol Chem. 2018 Apr 6;293(14):5259-5269. doi: 10.1074/jbc.RA117.000565). A prior paper also shows that Polθ-hel mutated in K121 also fails to promote ATP hydrolysis (Nature. 2015 Feb 12;518(7538):258-62. doi: 10.1038/nature14184).

Reviewer 1:

There are many examples that show ATP binding but not hydrolysis to be required for multimeric protein assembly (e.g. the proteasome <https://www.ncbi.nlm.nih.gov/pubmed/17018291>.) The high amounts of ATP needed for stability of the protein during purification could hint on ATP as a structural co-factor, rather than an enzymatic requirement. Thus, if the mutant retains ATP-binding, it may be sufficient for its biological function.

Authors response:

Because the K121 residue is essential for binding ATP (based on the crystal structure of Polθ-hel bound to ATP and biochemical and conservation studies of Polθ-hel and other SF2 helicases (Structure. 2015 Dec 1;23(12):2319-2330. doi: 10.1016/j.str.2015.10.014; J Biol Chem. 2018 Apr 6;293(14):5259-5269. doi: 10.1074/jbc.RA117.000565; *Curr Opin Struct Biol.* 2010 Jun;20(3):313-24. doi: 10.1016/j.sbi.2010.03.011; *Front Biosci* (Landmark Ed). 2012 Jun 1;17:2070-88.), we do not believe the nucleotide has a major role on the structural organization of Polθ. For example, the previous structural paper on Polθ-hel showed that multimerization of Polθ-hel is identical whether ATP is present or not (Structure. 2015 Dec 1;23(12):2319-2330. doi: 10.1016/j.str.2015.10.014). Our studies in the presented manuscript instead show that the central domain has a major influence on oligomerization of full-length Polθ. For example, replacement of the central domain with the short linker results in large oligomerization of the protein even without DNA (see Fig. 6A-D). The WT Polθ behaves mostly as monomers and dimers, but undergoes oligomerization when it binds DNA (see Fig. 6A,B). We note that ATP has been shown to act as a hydrotrope to increase the solubility of proteins (*Science* 19 May 2017: Vol. 356, Issue 6339, pp. 753-756 DOI: 10.1126/science.aaf6846), and this has now been shown for the proteome and demonstrated to increase the solubility of proteins with disordered domains like Polθ (*Nature Communications* (2019) 10:1155 <https://doi.org/10.1038/s41467-019-09107-y>). Thus, the inclusion of ATP in our protein preparation of Polθ likely increases solubility of the protein by acting as a hydrotrope rather than as a specific co-factor, especially since we purify PolθK121M the same way with ATP which does not bind to the mutated Walker A motif (i.e. K121M).

Reviewer 1:

While the authors nicely demonstrate that POLQ-hel promotes inter-molecular pairing by suppressing intra-molecular reactions, it would be insightful if the authors could discuss the requirement for a helicase domain in this process - even if the translocation activity requiring ATP hydrolysis is not required.

Authors response:

We again thank the reviewer for their insight and thorough review. We have thoroughly discussed the role of helicase binding upstream from Polθ-pol along the ssDNA overhang as the mechanism by which the helicase suppresses Polθ-pol intrastrand pairing. This is presented in the discussion section and is supported by structural models presented in Fig. 7E,G. The footprinting data in Fig. 5B,C directly support this model and the structural model presented in Fig. 5A also supports these findings as well.

Reviewer 1:

4) The interpretations with regards to the gel-like aggregates and the function of the POL-cen domain in this context leave many questions unanswered with alternative interpretations: The authors conclude that the major mechanistic function of Polθ-cen is to tether Polθ-hel to Polθ-pol (Fig.4B). However disordered regions play important roles for phase separation reactions, while “gelling” is associated so far with pathological protein aggregates, such as accumulated during

Alzheimers disease ([https://www.cell.com/trends/cell-biology/references/S0962-8924\(18\)30028-X](https://www.cell.com/trends/cell-biology/references/S0962-8924(18)30028-X), <https://www.biorxiv.org/content/10.1101/621714v1>) What is the evidence of gel-like aggregates versus fluid aggregates?

Authors response:

Here, we clarify our evidence supporting gel-like phase versus fluid phase separation. The specific pieces of evidence in support of a gel-like phase are: 1) the fact that Pol θ condensates do not resemble droplets and 2) the seemingly low density of the Pol θ condensates. The latter can be inferred from the observation that condensates are catalytically active and thus permeable to the DNA substrate. For example, we show that replacement of the central domain greatly increases large Pol $\theta\Delta$ cen complex formation with DNA, and these complexes are shown to be highly active and convert nearly 100% of the DNA substrates to end-joining products (see Fig. 6H-N). Concerning point 1, general arguments of thermodynamic stability imply that there is a surface tension associated to the phase boundary. As a consequence, liquid Pol θ condensates would be expected to adopt spherical shapes to minimize the surface area at any given volume. The highly irregular shapes observed for the condensates (i.e. Pol $\theta\Delta$ cen; Fig. 6K,M) suggest that Pol θ proteins do not diffuse freely inside them. Thus, Pol θ forms complexes that can be low-density (gel-like). In support of this, Pol $\theta\Delta$ cen which readily forms large complexes retains its catalytical capability during phase separation (Fig. 6H-N). Consistently, our theoretical model shows that the “dumbbell” geometry of Pol $\theta\Delta$ cen favors formation of a percolating network of pairwise interaction, thereby giving rise to a sponge-like aggregate that can accommodate DNA (see Fig. 7A-D). In summary, our hypothesis of a gel-phase rests on a logical deduction and on an unbiased, minimalist model of the phase separation process, as well as empirical images and video in real-time of the formation and movement of these Pol θ complexes which were modeled using the minimal end-joining protein Pol $\theta\Delta$ cen. Because the WT full-length Polq protein includes the large disordered central domain region, we utilized Pol $\theta\Delta$ cen to model Pol θ -Pol θ interactions as Pol $\theta\Delta$ cen is fully functional in MMEJ. Thus, we believe the WT Pol θ behaves similarly but appears less prone to forming such large complexes due to the regulatory role of the central domain which suppresses protein-protein interaction in the absence of DNA (see Fig. 7A-D and Fig. 6A-D).

We agree with the reviewer that some of our conclusions are preliminary and that further experiments will be needed to provide solid support to our hypothesis. We note, however, that formation of gel-like phases in cellular environments has been already discussed elsewhere (Harmon TS, Holehouse AS, Rosen MK, Pappu RV. Intrinsically disordered linkers determine the interplay between phase separation and gelation in multivalent proteins. *Elife*. 2017 Nov 1;6:e30294.; McManus, J.J., Charbonneau, P., Zaccarelli, E. and Asherie, N., 2016. The physics of protein self-assembly. *Current opinion in colloid & interface science*, 22, pp.73-79.) and reported to occur in cellular contexts (Cai, J., Townsend, J.P., Dodson, T.C., Heiney, P.A. and Sweeney, A.M., 2017. Eye patches: Protein assembly of index-gradient squid lenses. *Science*, 357(6351), pp.564-569.). We also note that we have obtained preliminary data showing the ability of endogenous Pol θ to form large complexes in ovarian cancer cells. However, this data requires much more analysis and further confirmation and characterization and will therefore be the focus of a follow-up study. Taken together, all these arguments make us reasonably confident that the Pol θ condensates are somewhat in-between a liquid and a solid phase.

Following the reviewer’s suggestion, we edited the manuscript to make more explicit the rational of our deduction and emphasize its tentative nature. The following statement was added to the main text:

“Another important feature is the shape of the complexes: in liquid-liquid phase-separation processes droplets assume a spherical shape in order to minimize surface area³¹. The highly irregular shape of the assemblies suggests that, rather than droplets, the proteins form extended percolating networks; moreover, the fact that they are catalytically active indicates that the assemblies are permeable to the solvent and, possibly, large solutes. Based on the computational and EM evidence, we surmise that Pol θ , and especially Pol $\theta\Delta$ cen, form large (>100 nm) gel-like complexes similarly to the ones reported

in refs^{32,33}, which accumulate DNA due to the high local concentration and accessibility of the polymerase and helicase domains.”

Reviewer 1:

Can the authors comment on the dissolution of these gel-like aggregates? Indeed, the authors show that in the absence of DNA, the POL-delta-cen dramatically increase aggregates compared to wild-type, which may be pathogenic, in a context where no pairing is wanted, e.g. in the absence of ssDNA overhangs. Thus, the conclusion that the cen domain solely acts as a linker may be premature.

Authors response:

We better clarified this point in the manuscript: the linker is likely playing more regulatory roles, such as suppressing Pol θ activity on short ssDNA and inhibiting oligomerization especially in the absence of DNA, possibly by masking protein-protein interactions along the Pol θ -pol and Pol θ -hel interfaces. For example, Pol θ -hel and Pol θ -pol are known to form tetramers and dimers, respectively, yet the WT Pol θ protein behaves mostly as monomers and dimers based on our EM and AFM data (see Fig. 6A,B). Replacing the central domain results in large complex formation even without DNA present (Fig. 6C,D,F,G). The addition of DNA allows WT Pol θ to form large complexes (Fig. 6A,B,E,F). Thus, it is conceivable that binding of DNA results in conformational change allowing the negatively charged central domain to move and uncover the protein-protein interaction interfaces, thereby promoting multimerization.

In response to the referee’s comments, we added this sentence in the discussion reflecting the role of the central domain:

“Thus, Pol θ -cen acts as a flexible linker, but also regulates the structure and function of Pol θ as discussed below.”

More detailed discussion of the structural and functional regulatory roles of the central domain are presented in the discussion section.

Reviewer 1:

E.g. aside the number of proteins in the oligomerized aggregates, is there any quality difference between the aggregates with and without DNA, and especially in the aggregates with and without the cen domain? Are the aggregates more or less fluid?

Authors response:

The microscopy data we have in the paper on the differences between the so-called aggregates of Pol θ and Pol $\theta\Delta$ cen with and without DNA are presented in Fig. 6. Specifically, replacement of the central domain results in a higher frequency of Pol $\theta\Delta$ cen:DNA complexes (Fig. 6I), and in the absence of DNA the Pol $\theta\Delta$ cen protein forms large intermolecular complexes even without DNA, whereas WT Pol θ requires DNA to form large complexes (see Fig. 6A-G). Relative rates of MMEJ by WT Pol θ and Pol $\theta\Delta$ cen proteins are similar. We did not perform a thorough statistical comparison between the two sets of images. A preliminary analysis suggests that, in absence of the linker, condensates tend to be more dense. However, we currently lack any quantitative insight into the issue, which we will further address in future studies.

Reviewer 1:

Alternatively, it seems that the authors show much data and insights on POLQ domain functions for MMEJ and pathway choice. The aggregation/ gelling is very interesting but perhaps could benefit from a more thorough, independent follow up study, and could be removed from this manuscript without taking away from the novelty and insight that Figs 1-5 provide.

Authors response:

We thank the reviewer for their suggestions and insight regarding the microscopy studies in our manuscript. In the case of the observations of WT Pol θ oligomerization being stimulated by DNA, we strongly feel that these data should remain in the paper. For example, prior reviews and reports on both the polymerase and helicase domain stress the ability of these separate domains to form multimers (tetramers for the helicase and dimers for the polymerase which is mainly based on crystallography). Even with my discussions with colleagues in this specific field, they reference the multimerization capabilities of the polymerase and helicase proteins as isolated enzymes. However, there is no available structural data for the full-length Pol θ protein. Therefore, our observations that WT full-length Pol θ behaves mostly as monomers and dimers in the absence of DNA but forms large complexes when DNA is present provides a major advancement for the field. Furthermore, the fact that the central domain greatly suppresses multimerization of Pol θ even when DNA is absent reveals a major regulatory role for this subdomain. It turns out that the Pol θ Δ cen protein happens to form very large complexes that are still very active and this led to the observations of gel-like properties rather than liquid-liquid phase separation in our opinion due to lack of observations of droplet formation. Taken together, we feel strongly that part of the major impact of this paper is the microscopy studies of the full-length Pol θ protein and mutants thereof since these data allow for a much more comprehensive structure function study of full-length Pol θ rather than just a biochemical analysis of this intriguing protein.

Reviewer 1:
Minor points:

The authors use long ssDNA substrates in the absence of RPA or other ssDNA stabilizing proteins to demonstrate that long ssDNA are needed for POLQ-MMEJ
Can the authors exclude that the ssDNA forms secondary structures and the ssDNA length requirement could reflect a need for dsDNA? This also could explain why full length POLQ is active on a p/tp construct even with short ssDNA overhang (Fig. 1F)

Authors response:

We have now added experimental data showing that double-strand DNA is not required for full-length Pol θ to perform MMEJ. The new data are presented in Supp Fig. 3I. Briefly, we tested Pol θ MMEJ with a new ssDNA substrate that is unable to form stable secondary structures based on the Integrated DNA Technologies oligo analysis tool. For example, the entire template upstream from the microhomology region (5'-CCCGGG-3') is composed of CA₃₂ repeats. The substrate length is 70 nt, and Pol θ performs highly efficient MMEJ of this substrate. In fact, there is very inefficient snap-back replication on this substrate compared to most other substrates tested which is likely due to the inability of Pol θ to form stable base-pairing between the 3' terminal end (5'-CCCGGG-3') and bases upstream along the same substrate (...CACACA...). Because this substrate is unable to form stable secondary structures based on its sequence composition, and the fact that Pol θ promotes efficient MMEJ of this substrate, we conclude that double-strand DNA due to secondary structure is not needed for Pol θ MMEJ. The following sentence was added to the text:

Further controls show that preventing secondary ssDNA structures enables efficient Pol θ MMEJ of long substrates (Supplementary Fig. 3I).

Reviewer 1:

If the ssDNA_x products in the gel are created from snap-back reactions, why are they not cleaved by XmaI, since it is also initiated by the same CCCGGG sequence. This could be clarified in the text.

Authors response:

Snap-back replication products were sequenced (see supplementary Fig. 4) and only one product appeared to result from snap-back occurring directly at the 3' terminus (i.e. between the GGG and CCC trinucleotide homologous sequences on the same strand). In this form of snap-back replication, the

resulting double-stranded product cannot be cleaved by XmaI since this enzyme requires the full double-strand CCCGGG/GGGCCC sequence, and not the half sequence (i.e. GGG/CCC) which can form from snap-back replication at the 3' terminus between the GGG and CCC microhomology within the same strand. Overall, the snap-back replication products result from transient base-pairing between the 3' terminal end with some microhomology within the middle region or 5' region of the ssDNA substrates- this is facilitated by Pol θ -pol which then extends the minimally paired 3' terminal end. This process is non-processive thus the enzyme can transiently pair and partially extend the substrate multiple times in a repetitive abortive pairing and extension process during snap-back replication.

Reviewer 1:

A lot of kinetic measurements are presented and it would be helpful to add the actual rate values, e.g. in ATPase figures of Fig 1G, or MMEJ rate of full length POLQ vs POLQ-delta-cen (fig 1N, 4K, and 4F) etc. This would help in comparing and in the actual description of the polymerase properties.

Authors response:

The initial rates of both ATP hydrolysis and MMEJ products formed which are calculated from the plotted data have now been included in the figure legends.

Reviewer 1:

What is the difference between Fig. 2J and A-C? In fact, it appears that the data was replotted judging from the error bars?

Authors response:

We again thank the referee for their thorough review. The summary data plotted in the original Fig. 2J was indeed taken from the 3 separate plots in Fig2A-C. This was meant to be completely transparent and show both the gel images next to the plotted data, and to present a full summary plot of all the data comparing the %MMEJ of substrates containing identical microhomology but with various lengths.

All the data are still included in Fig. 2, however, the panels in the figure have been re-ordered to reflect the order of the main text explanations which makes it easier for the reader to follow- this is in accordance with the referee's earlier request above to refer to figures and figure panels in chronological order in main text.

Reviewer 1:

Define "POL cen" on p4

Authors response:

Pol θ -cen and the other Pol θ subdomains are now defined in the second paragraph of the introduction section which has been shortened based on the referee's comments above.

Reviewer 1:

Define pssDNA

Authors response:

pssDNA (partial single-strand DNA) is defined in the text the first time it appears under the section titled, "Preventing Intrastrand Base-Pairing Stimulates MMEJ by Pol θ -Pol". pssDNA substrates are also illustrated in Figure 3.

We hope the referee agrees that the newly revised manuscript addresses all of the points raised during the review process and now represents a more clear and concise report on the molecular basis of MMEJ by full-length Pol θ .

Reviewer 2:

...I support its publication in Nature Communication, with the following minor suggestion for revision.

1). Full length polQ and polQ-dcen formed gel like oligomeric aggregates, which is a very intriguing proposal, and makes me wonder about the following to questions for the author to consider.

a). Could such a gel-like aggregates function as a storage site of polQ inside nucleus? Does polQ show some subcellular localization, in some sort of granular-structure in cells as a storage site for fast access when needed ?

Authors response:

We thank the referee for their thorough and fair review of our manuscript. We also believe the ability of Pol θ to form large active complexes is intriguing and that's why we explored some of these structural aspects of Pol θ in vitro using multiple microscopy methods since no structural information on full-length Pol θ is currently available in the literature. The idea that Pol θ forms large subcellular granular-structures is a very interesting possibility. In support of this idea, we have preliminary immunofluorescence data showing that Pol θ indeed forms large but infrequent complexes that appear to be in the nucleolus where other DNA repair proteins (i.e. WRN and BLM helicases) have been shown to localize to. Since these immunofluorescence data are still preliminary and need much more thorough analysis for confirmation and characterization, we will perform these studies as a full follow-up paper in the near future.

Reviewer 2:

b). Because of its ability to oligomerize on dsDNA, could polQ play a role in chromosome organization also associated with dsDNA break damaging and repair ?

Authors response:

This is another intriguing idea that needs to be further explored in follow-up studies. We note that our in vitro data investigated Pol θ binding to double-strand DNA in the absence of nucleosomes. Therefore, nucleosomes may influence the ability of Pol θ to form large complexes on double-strand DNA (i.e. away from DNA breaks) and this should be explored in future studies. We note that it has been very difficult for researchers in this field to image Pol θ in cells due to the inability to perform immunofluorescence. We now have identified a good antibody and ovarian cancer cell line that may be useful for imaging Pol θ localization in cells as noted above in our preliminary studies. We hope to follow up this study with assays designed to further analyze Pol θ localization in cells.

Reviewer 2:

2). What's the evidence of the central domain to be unstructured besides the disorder prediction? If really unstructured for such a long stretch (over 800 aa), it should not be stable (will be degraded inside the cells) or tangled with itself or other proteins. More likely it have extended structure to behave as an elongated and flexible domain. Maybe "flexible connector" or similar sort is a better term than unstructured domain for pol-cen?

Authors response:

We again thank the referee for their insight and fair review. We note that the central domain of Pol θ from invertebrates also lacks any predicted secondary domains, yet is significantly shorter (~250 aa) than the central domain from mammalian Polq (>800 aa). Due to the lack of predicted secondary domains within the central domains of Pol θ and lack of any conserved sequence, it has been hypothesized that this domain simply acts as a flexible linker, and our manuscript is the first direct evidence in support of this especially since replacing the central domain with a short flexible linker results in a protein (Pol θ Δ cen) with identical MMEJ activity. We, however, agree with the referee that this domain may indeed be associating with the other domains of Pol θ . For example, we have

emphasized the fact that replacing the central domain with the short flexible linker (Pol θ Δ cen) results in a protein that readily forms large protein-protein complexes even without DNA present. In contrast, WT Pol θ acts mostly as monomers and dimers without DNA, but oligomerizes upon binding to DNA. These data support the idea that the central domain although likely flexible prevents protein-protein intermolecular interactions in the absence of DNA. These data and points of discussion are presented in the discussion section and in Figures 6 and 7. In future studies we will examine replacement of the human central domain with those from invertebrates which also appear to be disordered based on sequence. We note that we are also pursuing cryoEM analysis of Pol θ to potentially elucidate the high resolution structure of this protein.

Reviewer 2:

3). Define hydrogen bonding potential? (similar to electrostatic potential or number of H bonds?)

Authors response:

We have now changed the text to simply “hydrogen bonding” instead of “hydrogen bonding potential”. This simply describes the ability of Pol θ to form DNA synapsis and end-joining via hydrogen bond formation between the opposing 3' overhangs via complementary bases (i.e. MMEJ).

Reviewer 2:

4). Fig. 1A, indicate the aa number at the beginning and end of polQ-dcen region.

Authors response:

The amino acid numbers have now been added to the figure 1A as requested.

Reviewer 2:

5). Fig. 6. Indicate ssDNA length used in legends for the different panels so that readers do not have to hunt that down in the main text.

Authors response:

We agree with the reviewer. We have now added the DNA lengths to the Figure 6 legends.

Reviewer 2:

6). Indicate the length of dsDNA used in the figure legend in S-Fig. 7.

Authors response:

The DNA lengths in Supplementary Figure 7 have now been defined in the legend.

Reviewer 2:

7). S-Fig. 7, in panels G-L, it appears polQ also bind dsDNA and form large aggregates. What's the binding affinity of polQ to dsDNA (for comparison to ssDNA binding). This affinity comparison may help understand the function of polQ in cells.

Authors response:

We thank the reviewer for this comment and we have noted above that nucleosomes may influence how Pol θ interacts with double-strand DNA away from DNA break ends where it likely primarily acts. We also note that Pol θ -pol shows poor DNA extension activity at double-strand DNA ends, but has efficient extension activity on 3' ssDNA overhangs (Elife. 2016 Jun 17;5. pii: e13740. doi: 10.7554/eLife.13740.). The Pol θ -hel, however, binds to both ssDNA and double-strand DNA and various types of DNA junctions (Structure. 2015 Dec 1;23(12):2319-2330. doi: 10.1016/j.str.2015.10.014.). Based on these prior findings, it is likely that Pol θ shows higher binding

affinity for 3' ssDNA ends compared to dsDNA. We also note that insofar Pol θ has been shown to exhibit more efficient cellular MMEJ activity on long (>45 nt) 3' ssDNA overhangs which further supports recruitment of this protein to DNA ends with 3' ssDNA overhangs. We are currently developing cellular assays to further investigate what type of DNA break substrates Pol θ works on, in particular what type of DNA ends accommodate its DNA repair activity.

Reviewer 2:

8). The model predicts the helicase binding to the 5' side of the ssDNA and pol to the 3' side. It follows that the 3' \leftrightarrow 5' translocation of helicase will allow the helicase to move further to the 5' end, away from pol at the 3'-end, fully extend the cen linker. This would help increase the binding to ssDNA if the initial binding through helicase domain happen to be at the 3'-end and pol domain not binding to ssDNA, as translocation from 3' \leftrightarrow 5' would allow binding of pol to the vacant 3' ssDNA.

Authors response:

We agree that this is an interesting point and we have thought about this possible mechanistic advantage. However, the fact that the Pol θ K121M mutant protein which lacks ATPase activity (and thus 3'-5' ssDNA translocation activity) acts in an identical manner to WT Pol θ in MMEJ (Fig. 4E,F), our data insofar indicate that the translocation function of the helicase has no obvious role in promoting MMEJ. We plan to perform a follow-up study on investigating whether Pol θ ATPase activity has a role in dissociating RAD51 or other proteins from ssDNA.

Reviewer 3:

...The report is well written, the data are fully documented, and the conclusions follow from the data. The report will appeal to a large audience, and I fully expect will garner many citations. In overview, I have no major comments and believe the report is of wide interest in the major fields of repair, recombination and replication. Minor comments that the authors may consider are listed below.

Minor Comments:

1) Given the expected orientation of Pol and helicase, modeled in Fig. 5, would the authors have examined whether the helicase activity is needed in the presence of ssDNA binding factors like Rad51 or RPA? This might be part of the authors future plans, but if this has been done already, it would help the reader understand if the helicase plays a role in this reaction. For example, the way it reads now – it seems that the helicase might be involved in a distinct molecular pathway.

Authors response:

We are grateful to the reviewer for their fair and thorough review. Regarding the helicase, there was a prior report indicating that the isolated helicase domain can dissociate RAD51 from ssDNA. However, many in the field believe this biochemical activity needs to be more thoroughly vetted. Our own prior studies have also shown that the isolated helicase domain can dissociate RPA from ssDNA in an ATP-dependent manner (J Biol Chem. 2018 Apr 6;293(14):5259-5269. doi: 10.1074/jbc.RA117.000565). Our future studies aim to fully examine these distinct biochemical activities of the full-length Pol θ protein. A prior report on Pol θ in *Drosophila* provided strong evidence that the helicase ATPase activity is important for an unknown function in interstrand crosslink repair but not MMEJ (PLoS Genet. 2017 May 25;13(5):e1006813. doi: 10.1371/journal.pgen.1006813.). This potential activity in mammalian cells however has not been investigated but remains an interesting possibility. We expect future studies to uncover relevant enzymatic activities of the Pol θ helicase domain in addition to its ability to promote MMEJ via ssDNA binding as presented herein.

Reviewer 3:

2) If the helicase domain were to mainly be a dsDNA translocase instead of a regular bona fide

helicase, it might translocate along dsDNA for the purpose of rapidly targeting the Pol domain to an end, while at the same time binding ssDNA of long 3' overhangs to suppress intramolecular snapback. Have the authors examined the FL Pol theta for helicase activity in simple helicase assays?

Authors response:

We have fully examined the helicase activities of the isolated helicase domain in a recent JBC paper (J Biol Chem. 2018 Apr 6;293(14):5259-5269. doi: 10.1074/jbc.RA117.000565). We found that Polθ-hel translocates along ssDNA in a 3'-5' direction in an ATP-dependent manner and is capable of non-processive DNA unwinding on short substrates. The helicase also can unwind short substrates modeled after replication forks. These previous studies demonstrate for the first time that Polθ-hel exhibits weak ATP-hydrolysis dependent DNA unwinding activity and acts similarly to related Hel308 type helicases. In future studies, we plan to fully examine the helicase activities of full-length Polθ.

Reviewer 3:

3) the authors might consider in the future to try experiments comparing the ATPase mutant vs wt FL Pol theta on substrates containing different lengths of dsDNA? If the helicase domain is mainly for dsDNA translocase action (e.g. like SMARCAL1 etc), the ATP proficient enzyme may enable more rapid binding/location of Pol theta to the 3' ssDNA end, as assayed using the restriction enzyme protection assay developed in this report.

Authors response:

We again thank the referee for their biochemical insight. We have tried in previous studies to identify dsDNA translocase activity by the isolated Polθ-hel to no avail. In future studies, we plan to fully examine the helicase activities of the full-length Polθ protein on various ssDNA and dsDNA substrates, and will try the assays suggested by the reviewer.

Reviewer 3:

4) The authors may want to consider (in future studies, not this one) cross-linking mass spectrometry of FL Pol theta +/- ssDNA, to map how extensive of an interface, if any, exists between the helicase and Pol domains. This would be informative to see if the helicase-Pol communication is direct, or indirect, and whether there are significant changes upon binding to a long 3' ssDNA overhang.

Authors response:

We agree that cross-linking assays would provide the most detailed binding mechanism by the full-length protein. We plan to perform these type of assays in future studies to further detail the configuration of the full-length Polθ on different DNA substrates. For example, how Polθ binds and acts on 3' ssDNA terminal ends may be elucidated by these experiments.

Reviewer 3:

5) The Pol theta lacking the cen region appears as capable of MMEJ as the full length protein. This suggests, as the authors note, that the cen domain is not essential. However, it would be informative to test the reaction in a NaCl titration, comparing wt Pol theta with Pol delta cen, to determine if they still contain similar activity when challenged by ionic strength (or by competitor ssDNA).

Authors response:

We again thank the reviewer for their insight. We have tried the MMEJ assay in the presence of excess ssDNA substrate immediately after initiating the reaction in Supp. Fig. 3C. The results shows that the excess ssDNA does indeed trap the protein. However, the proteins that have already engaged in initial extension of the 3' terminal ends immediately following the end pairing step appeared to be resistant to the ssDNA excess trap. Hence, these data which are briefly described in the manuscript show that the MMEJ reaction is processive only after the initial ssDNA pairing step when the Pol domain can initiate extension. We have also performed MMEJ assays testing WT Polθ vs PolθΔcen in the presence of

increasing ionic strength and did not find significant differences between their respective activities. These data therefore were not included in the manuscript.

Reviewer 3:

6) It became a bit confusing to read that the cen region is not required for MMEJ, yet to also read that the central region is demonstrated to be required to restrict the MMEJ activity to long 3' overhangs, and down regulates MMEJ of short 3' overhangs. Maybe it was just me. But if other reviewers make the same comment, perhaps a bit of clarification on what the authors are thinking regarding the cen region.

Authors response:

We again thank the referee for their careful reading of the manuscript. We agree that the central domain has multiple functions and this may make the findings somewhat complex at times. We tried to describe the multiple functions of the central domain clearly and concisely in the discussion section with simple images in Fig. 7. We hope the reader will now understand that the central domain is necessary for linking the polymerase and helicase domains, but also suppresses the polymerase domain on short ssDNA substrates and therefore has an autoinhibitory role on short ssDNA substrates.

Reviewer 3:

7) The following comment does not need an experiment to be performed for the current publication, but I only mention it out of my own curiosity.

Regarding the multimeric status of Pol theta in the reaction, and the question of whether large multimers of the enzyme are needed, would this assay be amenable to the separation of the reaction into two steps such that only one end of each of two substrates with micro homology is preincubated with Pol theta, and then to remove excess Pol theta (eg gel filtration) before mixing and examining the MMEJ reaction? Likewise, to see if preinc of both substrates, separately (and gel filtered separately) is sufficient for MMEJ. While this may or may not change the multimerization conclusion, it might determine if the multimerization leading to active MMEJ requires both ends to be simultaneously present in the same reaction to form the proper network for joining.

Authors response:

This is indeed an interesting assay for future studies. The fact that Pol θ requires two long substrates to perform MMEJ suggests that at least two molecules of the enzyme are needed for this reaction. For example, since WT Pol θ can only perform MMEJ on long ssDNA substrates, this suggests that the enzyme becomes activated for MMEJ specifically on long ssDNA. Indeed, we found that WT Pol θ cannot perform MMEJ between a long and short ssDNA, but can only perform MMEJ between two long ssDNA substrates. These data are presented in Fig. 5G. We hope to solve the cryo-EM structure of Pol θ as apoenzyme and on ssDNA substrates in the future which should provide the most in depth molecular insight into how this protein performs ssDNA synapsis.

REVIEWERS' COMMENTS:

Reviewer #1 (Remarks to the Author):

All comments have been satisfactorily addressed.

Reviewer #2 (Remarks to the Author):

This reviewer is satisfied with the author response and revision.